# Counterregulation of cAMP-directed kinase activities controls ciliogenesis

Monia Porpora[1], Simona Sauchella[1], Laura Rinaldi[1], Rossella Delle Donne[1], Maria Sepe[1], Omar Torres-Quesada[2], Daniela Intartaglia [3], Corrado Garbi[1], Luigi Insabato[4], Margherita Santoriello[1], Verena A. Bachmann[2], Matthis Synofzik[5], Herbert H. Lindner[6], Ivan Conte[3], Eduard Stefan [2] & Antonio Feliciello [1]

The primary cilium emanates from the cell surface of growth-arrested cells and plays a central role in vertebrate development and tissue homeostasis. The mechanisms that control ciliogenesis have been extensively explored. However, the intersection between GPCR signaling and the ubiquitin pathway in the control of cilium stability are unknown. Here we observe that cAMP elevation promotes cilia resorption. At centriolar satellites, we identify a multimeric complex nucleated by PCM1 that includes two kinases, NEK10 and PKA, and the E3 ubiquitin ligase CHIP. We show that NEK10 is essential for ciliogenesis in mammals and for the development of medaka fish. PKA phosphorylation primes NEK10 for CHIP-mediated ubiquitination and proteolysis resulting in cilia resorption. Disarrangement of this control mechanism occurs in proliferative and genetic disorders. These findings unveil a pericentriolar kinase signalosome that efficiently links the cAMP cascade with the ubiquitin-proteasome system, thereby controlling essential aspects of ciliogenesis.

[1] Department of Molecular Medicine and Medical Biotechnologies, University 'Federico II', Naples 80131, Italy. [2] Institute of Biochemistry and Center for Molecular Biosciences Innsbruck, University of Innsbruck, A-6020 Innsbruck, Austria. [3] Telethon Institute of Genetics and Medicine, Pozzuoli (Naples) 80078, Italy. [4] Department of Advanced Biomedical Sciences, University Federico II, Naples 80131, Italy. [5] Department of Neurodegeneration, Hertie Institute for Clinical Brain Research (HIH), University of Tübingen and German Center for Neurodegenerative Diseases (DZNE), 72076 Tübingen, Germany. [6] Division of Clinical Biochemistry, Biocenter Medical University of Innsbruck, Innrain 80-82, A-6020 Innsbruck, Austria. These authors contributed equally: Monia Porpora, Simona Sauchella. Correspondence and requests for materials should be addressed to A.F. (email: feliciel@unina.it)

Primary cilia are sensory organelles that receive, integrate, and transmit a variety of extracellular signals to intracellular compartments. Receptors, ion channels, transporter proteins, scaffolds, and effector proteins localize and function at ciliary compartments. The primary cilium focuses signal transmission and contributes to cell homeostasis during development and tissue remodeling[1,2]. Recent findings support a key role of the primary cilium in important aspects of vertebrate development and tissue homeostasis. Altered ciliogenesis or dysfunctional cilia cause ciliopathies that have been causally linked to a wide range of genetic and proliferative diseases[3]. Therefore, understanding of the basic and conserved mechanism of ciliogenesis or cilium removal will expose new avenues for pharmacological targeting of such disorders.

Primary cilia extend from the basal body, which is derived from the mother centriole of the centrosome and consists of an axoneme formed by nine doublet microtubules surrounded by the ciliary membrane. Cilium assembly is induced when cells deprived of mitogens leave the cell cycle. This process is initiated by the docking of ciliary vesicles at the distal site of the basal body. The growth of axonemal microtubules and subsequent fusion of the nascent cilium with plasma membrane culminates in the formation of mature cilia[1]. A wide array of pericentriolar proteins have been identified as major regulators of cilia assembly, growth, and maintenance[4]. The pericentriolar matrix protein 1 (PCM1), a central component of centriolar satellites, is localized within the electron dense granules scattered around centrosomes.

PCM1 acts as scaffolding platform to organize centrosomal and pericentriolar proteins that are implicated in the spatiotemporal dynamics of both centrioles and the microtubule network[5]. The central role of PCM1 in ciliogenesis has been described[6]. Regulators, effectors, and components of the ciliary compartment form macromolecular complexes with PCM1. Accordingly, depletion of PCM1 leads to delocalization of its pericentriolar and ciliary partners and to a concomitant loss-of-primary cilia[7,8]. PCM1 is also a target of the ubiquitin-proteasome system (UPS). In growing cells, ubiquitylation of PCM1, AZI1, and CEP290 by the E3 ligase MIB1 suppresses primary cilium formation. Under stress conditions, inactivation of MIB1 by stress kinases abolishes AZI1, PCM1, and CEP290 ubiquitylation and promotes ciliogenesis in proliferating cells[9–11].

Components of the cAMP cascade, such as G-protein coupled receptors (GPCRs), adenylate cyclases (ACs), and phosphodiesterases (PDEs) are central signaling units that act on the primary cilium and are functionally implicated in critical aspects of cilium formation and signaling[12–14]. Proteomic screening and in situ immunolocalization studies identified cAMP-dependent protein kinase A (PKA) holoenzyme as a main component of the ciliary compartment. Localization of PKA to the cilium and its cAMP-dependent spatiotemporal activation is important for antagonizing Hedgehog-initiated signaling, which is essential for normal embryonic development[13,15,16]. Recently, the orphan GPCR Gpr161, known to be involved in cAMP and hedgehog signaling has been identified as a scaffolding protein (A kinase anchoring

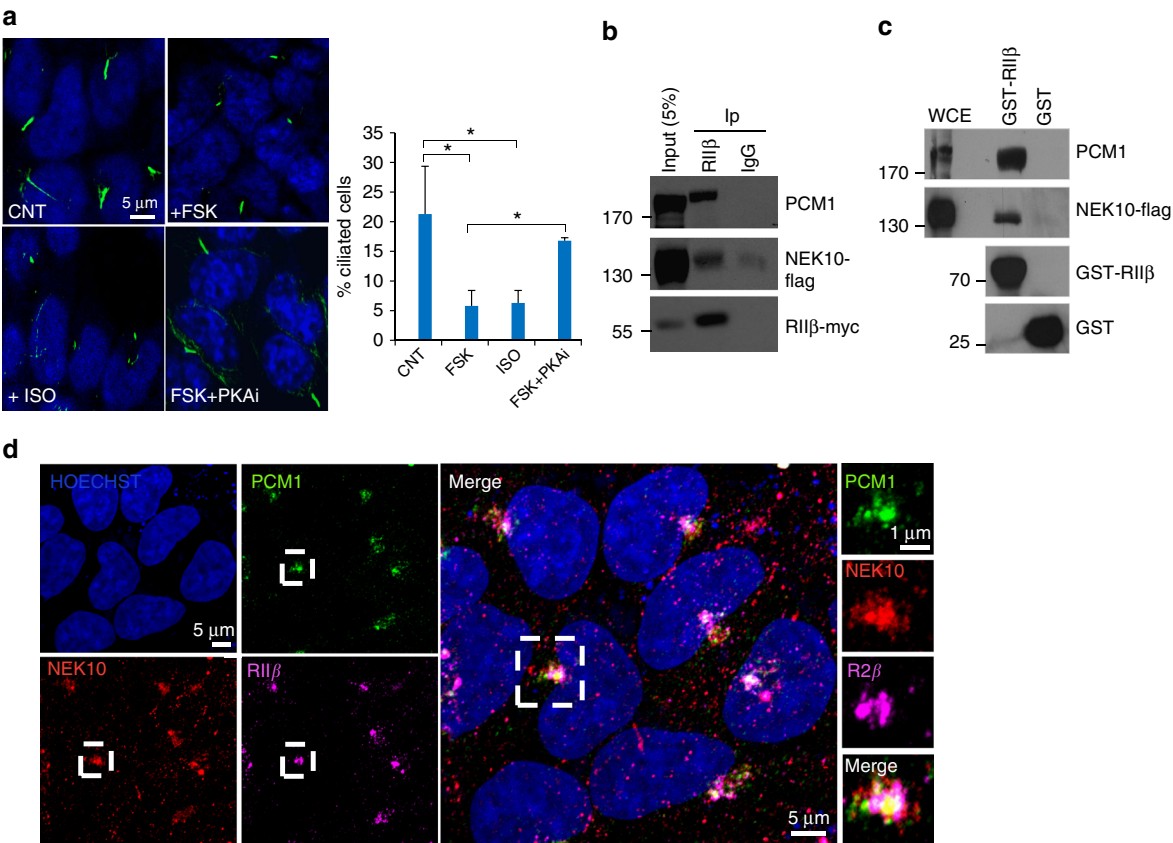

**Fig. 1** Assembly of a pericentriolar kinase complex. **a** HEK293 cells were serum-deprived for 36 h and then left untreated (CNT) or stimulated for 3 h with isoproterenol (Iso) or forskolin (FSK) and doubly stained for acetylated tubulin and Draq5. Cumulative data from five independent experiments are shown. *$p < 0.05$ (Student's $t$-test). Where indicated, the cells were pretreated for 30 min with the PKA inhibitor (PKAi) H89 (5 μM) before stimulation. **b** RIIβ was immunopurified from whole-cell extracts (WCE). IgG was used as control. The precipitates and lysates were immunoblotted with anti-PCM1 and anti-RIIβ antibodies. **c** Lysates of NEK10-flag expressing cells were subjected to pull down assays with purified GST or GST–RIIβ fusion. The precipitates and lysates were immunoblotted with anti-PCM1, anti-flag and anti-RIIβ. **d** HEK293 cells were fixed and immunostained for endogenous PCM1, NEK10, and RIIβ. A merge composite of the three signals and magnification of a selected area are shown (right panels)

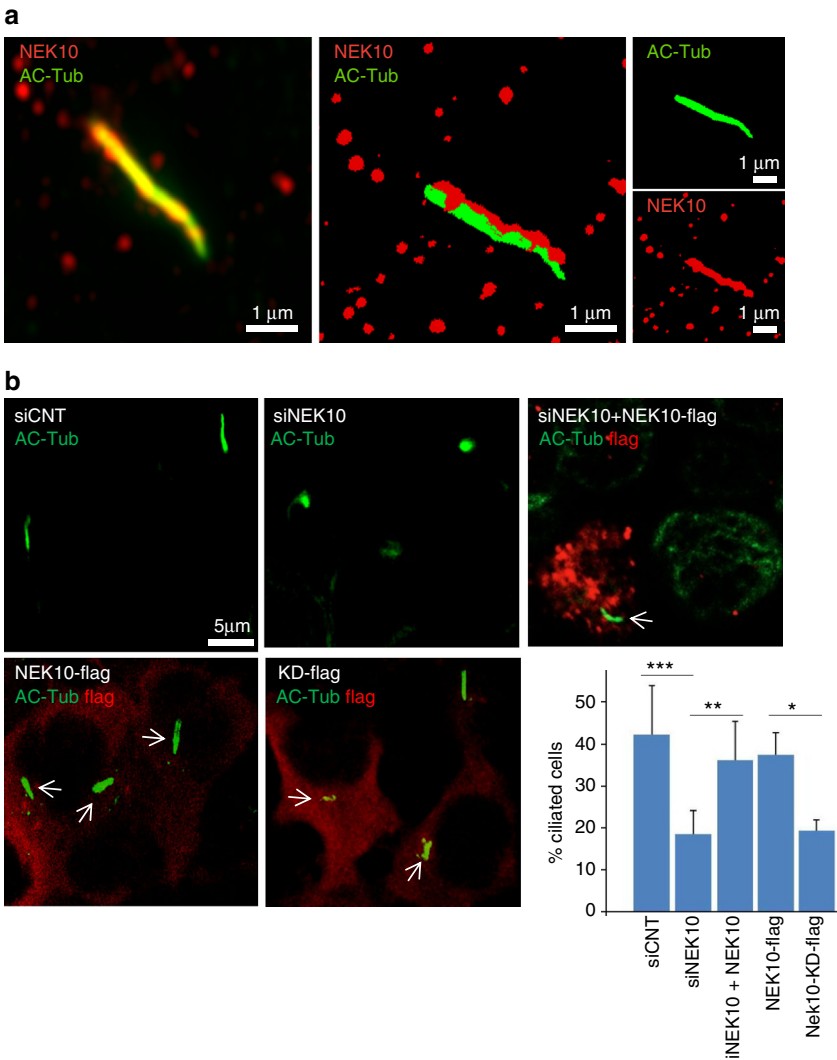

**Fig. 2** NEK10 localization and regulation of primary cilium formation. **a** Serum-deprived HEK293 cells were immunostained for NEK10 and acetylated tubulin and analyzed by confocal microscope equipped with Airyscan superResolution imaging module. A merge composite 2D and 3D of the signals is shown. **b** HEK293 cells were transiently transfected with control (siCNT) or with siRNAs targeting NEK10 (siNEK10), serum deprived for 36 h, formalin-fixed and immunostained for acetylated tubulin. Where indicated, NEK10-flag vector (either wild type or kinase dead, KD) was included in the siRNAs transfection mixture. NEK10 expression was visualized with anti-flag antibody. Arrows indicate the localtion of the cilium in cells expressing flag-tagged NEK10. Cumulative data from three independent experiments are shown (lower right panel). For each group a minimum of 100 cells/experiment was averaged. $*p < 0.05$; $**p < 0.01$; $***p < 0.001$ (Student's $t$-test)

protein; AKAP) for recruiting PKA to the primary cilium[13,17]. Such scaffold mediated targeting of PKA holoenzymes in proximity of its substrates, optimizes the biological responses to hormone stimulation[18–20]. The question arises if other macromolecular PKA complexes at the base of cilium are involved in cilium formation. Delocalization of PKA from the cilium profoundly impacts on downstream developmental pathways, suggesting distinct PKA signalosomes that locally control cAMP messages not only at the organelle but also for its formation[12]. However, the composition of the macromolecular PKA complexes at the basis of this ciliary compartment is still unknown. Moreover, the link between the cAMP pathway and the control of cilium resorption in hormone-stimulated cells is still undefined. Here we fill this gap by identifying PCM1 as an interlinking platform for cAMP-sensing and kinase (NEK10 and PKA) counterregulation, which is actually relevant for ciliogenesis.

We found that NEK10 is required for cilia biogenesis, both in mammals and lower vertebrates (i.e., medaka fish). PKA

phosphorylation induces ubiquitination and proteolysis of NEK10 by the co-assembled E3 ligase CHIP. We present evidence that the loss-of-NEK10 promotes cilium resorption. This UPS and kinase-involved control mechanism for cilium dynamics might be deregulated in specific human cancers and genetic disorders.

## Results

**PCM1 targets PKA and NEK10 to the centriolar satellites.** Activation of PKA by cAMP agonists at the primary cilium is functionally linked to developmental pathways[12]. However, if and how PKA regulates the stability of primary cilium in growth-arrested cells remained largely elusive. We therefore tested the impact of PKA activation on cilium stability. We used an experimental model where the formation of primary cilia can be induced by serum deprivation[21]. Cilia are visualized by immunofluorescence using an antibody directed against acetylated tubulin, a modified form of tubulin that specifically accumulates at primary cilia[22]. As shown in Fig. 1a, serum deprivation for 36 h

promoted the formation of primary cilia (CNT). Interestingly, activation of adenylate cyclase by the diterpene forskolin (FSK), stimulation of beta-adrenergic receptor pathways (bAR) by iso-proterenol (ISO) or serum readdition induced a rapid resorption of primary cilia (Fig. 1a, Supplementary Fig. 1a-b). Serum resti-mulation significantly elevated the intracellular levels of cAMP in serum deprived cells, as did forskolin or isoproterenol that maximaly activate it (Supplementary Fig. 1c-d). Serum readdition also increased phosphorylation of PKA substrates (Supplemen-tary Fig. 1e). Treatment with drugs did not affect cell viability, since readdition of serum to FSK-treated cells restored cell pro-liferation (Supplementary Fig. 1f). Moreover, the effects of cAMP stimulation were mediated by PKA, since inhibiting the kinase by a pharmacological inhibitor (Fig. 1a) or by PKI (Supplementary Fig. 2) prevented cilium resorption induced by foskolin treat-ment. This finding suggested a control mechanism of cilia sta-bility in starved cells mediated by the cAMP pathway.

Large-scale proteomic analyses identified PKA as a component of a macromolecular complex nucleated by PCM1[23]. Moreover, we identified the Nima-related Kinase 10 (NEK10) in the unprocessed proteomics dataset of PKA interactors[13]. NEK10 is a member of the NEK family kinases involved in microtubule dynamics, cell cycle progression and ciliogenesis[24]. Based on these observations, we investigated how PKA intersects with NEK10 pathway to control primary cilium stability. First, we asked if PKA and NEK10 can be assembled within the same complex by PCM1. Co-immunoprecipitation assays (Fig. 1b) and GST-pull-down experiments (Fig. 1c) revealed a trimeric complex composed of PCM1, NEK10, and RIIβ. Furthermore, immunos-taining analysis revealed that PCM1, NEK10, and RIIβ signals partially co-localized at pericentriolal region, supporting the notion that the three proteins can be present within the same intracellular compartment (Fig. 1d).

Next, we analyzed the binary interaction between PCM1 and both kinases (PKA and NEK10). Co-immunoprecipitation assays confirmed that PCM1 and NEK10 form a stable complex in lysates (Supplementary Fig. 3a). To identify the domain of PCM1 that mediates the NEK10 binding, we generated a series of deletion mutants and tested their ability to interact with the kinase. Co-immunoprecipitation assays demonstrated that residues 941–1207 of PCM1 mediate the interaction with NEK10 (Supplementary Fig. 3b-d). Similarly, we studied the binding between PCM1 and PKA. Co-immunoprecipitation experiments of endogenous proteins confirmed the presence of PCM1 and PKA within the same complex (Supplementary Fig. 4). Additionally, double immunostaining for endogenous RIIβ subunit and PCM1 confirmed that a significant fraction of PKA co-localized with PCM1 (Supplementary Fig. 5a). Genetic knockdown of PCM1 reduced the intensity of pericentriolar RIIβ staining, suggesting that PCM1 was, at least in part, required for PKA localization at centriolar satellites (Supplementary Fig. 5b). Despite the evidence that PKA and PCM1 form a stable complex in cAMP precipitation studies[25] (Supplementary Fig. 6), we could not confirm binary interactions using in vitro binding assays with recombinant proteins. These findings are consistent with the notion that the interaction between these proteins is indirect.

**Essential role of NEK10 in ciliogenesis.** Members of the NEK family have been implicated in various aspects of microtubule dynamics and ciliogenesis[26–29]. NEK10 localizes at centriolar satellites. We therefore tested whether localization was involved in regulating primary cilium formation. Immunostaining analysis confirmed that a significant fraction of NEK10 is, indeed, loca-lized at cilia where it appeared uniformly distributed along the axoneme (Fig. 2a and Supplementary Movie 1). Note that in some cells a NEK10 staining at the base of the cilium could be detected, suggesting that localization of this kinase at ciliary compartment is a dynamically regulated mechanism. Given its localization at cilia, we tested whether NEK10 is required for ciliogenesis. As shown in Fig. 2b (upper panels) and Supplementary Fig. 7a, genetic silencing of NEK10 reduced the number of ciliated cells. Re-introduction of an exogenous NEK10 that cannot be targeted by the siRNA rescued cilia in NEK10-depleted cells. The con-tribution of NEK10 in ciliogenesis was also analyzed by inter-fering with its kinase activity. Thus, expression of a kinase-dead mutant of NEK10 (NEK10-KD) carrying the K548R mutation[30] significantly reduced the number of ciliated cells (Fig. 2b, lower panels). To provide insights into how Nek10 regulates ciliation, we performed a proteome analysis of NEK10 complexes in cAMP

**Table 1 Selected NEK10-associated proteins identified by mass spectrometry analysis. Cells transiently expressing NEK10-Flag WT or the mutant T812A were subjected to immuno-precipitations (IPs) using Flag antibody (IPs were done in the presence of serum following 10 μM Forskolin treatment for 15 min). IPs performed with anti-GFP antibodies served as control. A selection of enriched and high-scoring proteins identified in two independent experiments are presented**

| Accession # | Gene name | Description | Score[a] |
|---|---|---|---|
| Q5T9A4 | ATD3B | ATPase family AAA domain-containing protein 3B | 279.72 |
| Q9NVI7 | ATD3A | ATPase family AAA domain-containing protein 3A | 177.73 |
| Q5SW79 | CE170 | Centrosomal protein of 170 kDa | 79.81 |
| Q13263 | TIF1B | Transcription intermediary factor 1-beta | 46.99 |
| Q96P47 | AGAP3 | Arf-GAP with GTPase, ANK repeat and PH domain-containing protein 3 | 40.07 |
| Q9UPQ3 | AGAP1 | Arf-GAP with GTPase, ANK repeat and PH domain-containing protein 1 | 38.88 |
| P12236 | ADT3 | ADP/ATP translocase 3 | 26 |
| Q6PID8 | KLD10 | Kelch domain-containing protein 10 | 22.44 |
| P49674 | KC1E | Casein kinase I isoform epsilon | 17.35 |
| Q99755 | PI51A | Phosphatidylinositol 4-phosphate 5-kinase type-1 alpha | 15.94 |
| Q5JTW2 | CEP78 | Centrosomal protein of 78 kDa | 15.9 |
| O00139 | KIF2A | Kinesin-like protein KIF2A | 14.37 |
| P27348 | 1433T | 14-3-3 protein theta | 11.62 |
| Q3KQU3 | MA7D1 | MAP7 domain-containing protein 1 | 10.92 |
| Q9HDC5 | JPH1 | Junctophilin-1 | 9.56 |
| P62258 | 1433E | 14-3-3 protein epsilon | 9.51 |
| P46940 | IQGA1 | Ras GTPase-activating-like protein IQGAP1 | 7.28 |

[a]Mascot score: protein score reflects the combined scores of all observed mass spectra that can be matched to amino acid sequences within that protein. A higher score indicates a more confident match

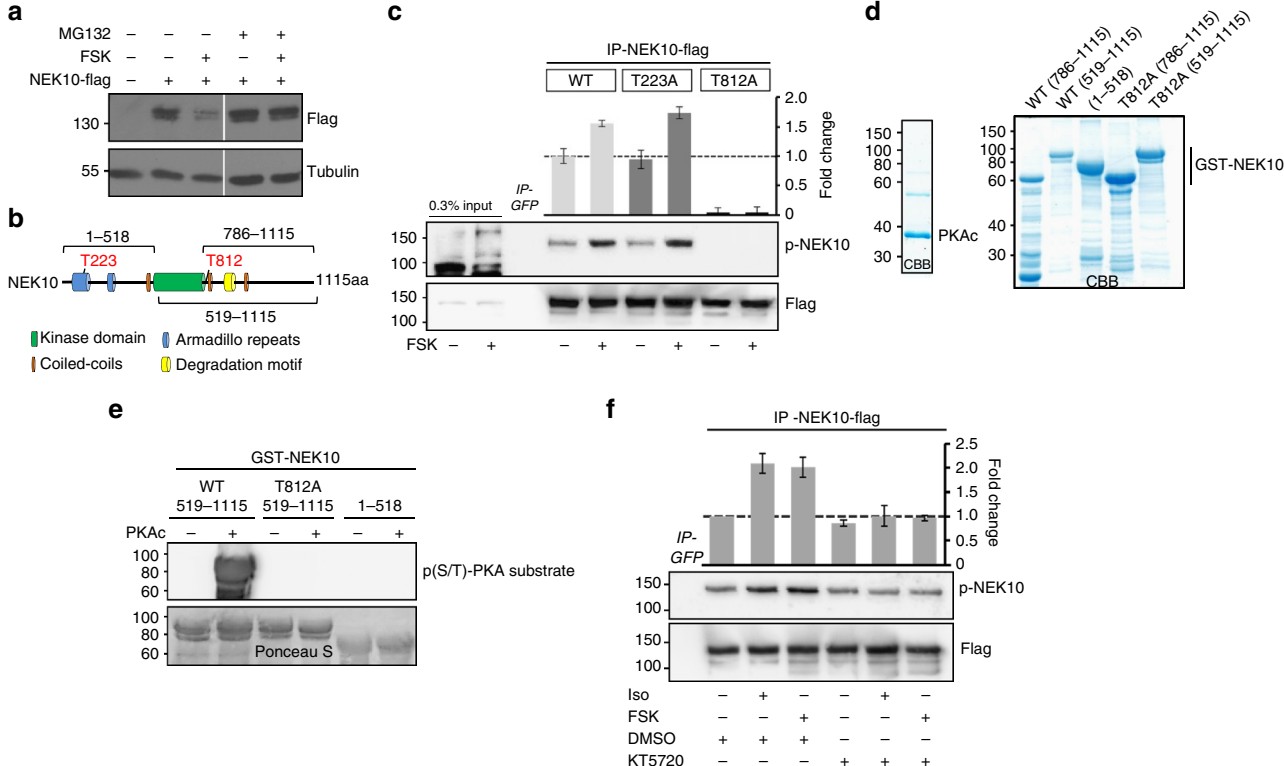

**Fig. 3** PKA phosphorylates NEK10 in vivo and in vitro. **a** Transfected HEK293 cells were left untreated or stimulated with forskolin for 60 min. Where indicated, MG132 (20 μM) was added to the medium. Lysates were immunoblotted for flag and tubulin. **b** The domain organization of NEK10 and the position of T223 and T812 are indicated. Composition of truncated NEK10 expression constructs are shown. **c** Cells transfected with either wild-type NEK10-flag or with NEK10-flag mutants (T223A-Flag and T812A-Flag) were left untreated or stimulated with FSK (15 min). NEK10 was immunopurified with anti-flag antibodies. The precipitates were immunoblotted with anti-flag and with anti-phospho-(K/R)(K/R)X(S*/T*) specific antibodies. The quantification is shown from $n = 4$ independent experiments (±SEM). **d** Commassie brilliant blue (CBB) stained gels of recombinant GST hybrid NEK10 proteins composed of the phosphorylation site mutation T812A and distinct parts of NEK10 are shown. **e** Recombinant and GST-fused NEK10 proteins were used as substrates for in vitro phosphorylation assays using recombinant and hexa-histidine-tagged PKAc. GST-immobilized fusion proteins were analyzed in immunoblotting experiments with an anti-phospho-(K/R)(K/R)X(S*/T*) specific antibody. Shown is a representative result from $n = 3$ independent experiments. **f** HEK293 cells transiently expressing NEK10-flag were treated with KT5720 (5 μM; 60 min) and then stimulated with forskolin (20 μM; 15 min) or isoproterenol (100 nM; 15 min). Cells were lysed and NEK10-flag was immunoprecipitated using anti-flag antibodies. Immunoblotting was performed with anti-flag and with anti-phospho-(K/R)(K/R)X(S*/T*) specific antibodies. The quantification from $n = 4$ independent experiments (±SEM) is shown

stimulated cells. Table 1 presents a selection of enriched and high-scoring proteins. In the IPs from the overexpressed HEK293 cell system, we could not detect major differences of interactions between wt and the T812A mutant. Centrosomal proteins and members of cytoskeleton-associated molecular motor proteins show a high-score probability as NEK10 interactors, suggesting a role for this kinase in cytoskeletal events taking place at centriolar satellites. However, further work is required to confirm this hypothesis and characterize the relevant NEK10 substrates involved in ciliogenesis.

**PKA phosphorylation primes NEK10 for proteolysis.** To gain further insight into the mechanism of cAMP action on cilia formation, we monitored NEK10 levels in cells stimulated with FSK. Figure 3a shows that FSK treatment reduced the levels of NEK10. Pre-treating the cells with MG132, a proteasome inhibitor, restored NEK10 levels in the presence of FSK (Fig. 3a). Importantly, FSK-induced loss-of-NEK10 was reproduced in cells pretreated with cycloheximide, an inhibitor of translation, supporting the concept that cAMP acts through the UPS to control NEK10 stability (Supplementary Fig. 7b). The data above indicate that, in response to cAMP stimulation, NEK10 undergoes

proteasomal degradation. We assume that PKA phosphorylation primes NEK10 for proteolysis. Primary sequence analysis of NEK10 predicts two conserved PKA phosphorylation sites (T223 and T812) (Fig. 3b and Supplementary Fig. 8). To ask if phosphorylation of one or both of these sites renders NEK10 susceptible to proteolysis, we generated mutant forms of NEK10 using a site-directed mutagenesis approach to substitute either T223 or T812 with alanine. We tested our hypothesis by analyzing the phosphorylation status of affinity-isolated NEK10 with a PKA substrate antibody. In contrast to phosphorylation of the wild type and T223A NEK10 mutant, the substitution of T812A abolished both basal and FSK-induced NEK10 phosphorylation (Fig. 3c). To prove that PKA directly phosphorylates NEK10, we purified a series of recombinant and GST-fused NEK10 hybrid proteins spanning distinct NEK10 domains from bacterial BL21 cell lysates (Fig. 3d). The purified polypeptides were subjected to in vitro phosphorylation assays using recombinant his-tagged PKAc. As shown in Fig. 3e and Supplementary Fig. 9, the C-terminal fragments of NEK10 were efficiently phosphorylated by PKAc. In contrast, the T812A mutation completely prevented phosphorylation of the NEK10 protein fragments by PKAc. To confirm that indeed PKA mediates NEK10 phosphorylation in cells, we pretreated cells with the PKA

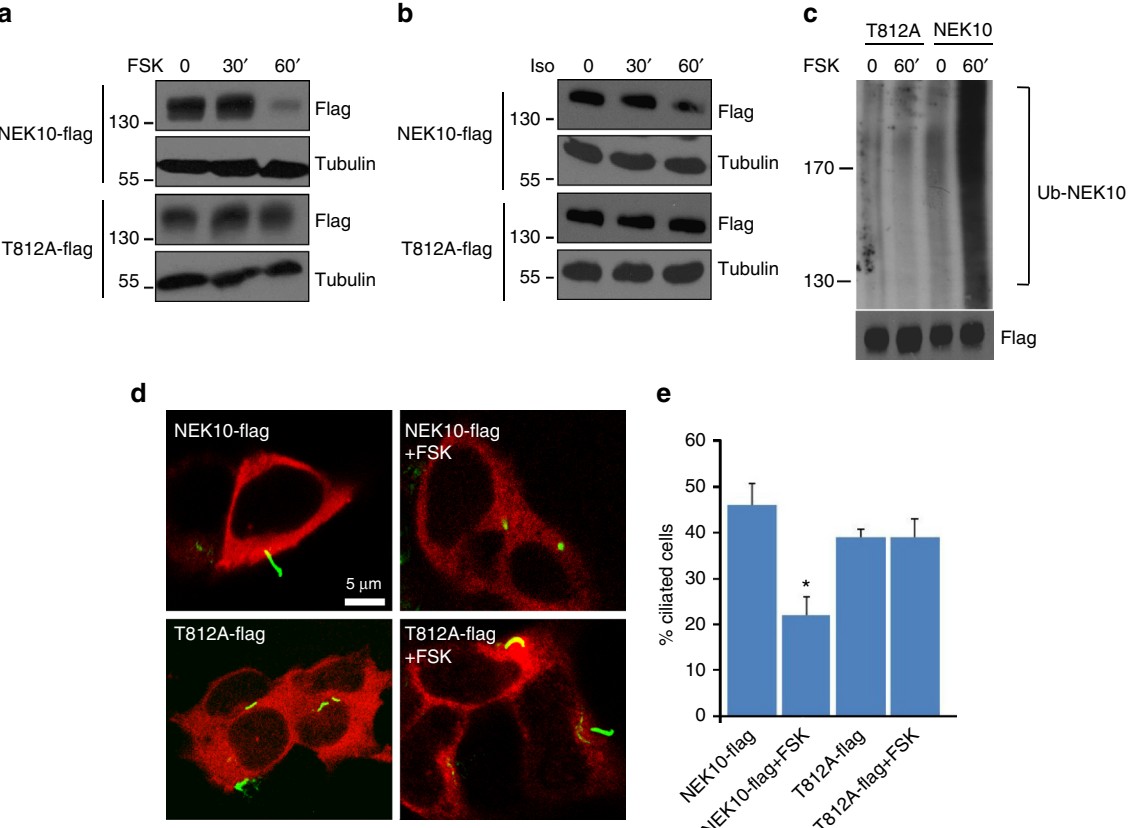

**Fig. 4** PKA phosphorylation induces NEK10 proteolysis and cilia resorption. **a–b** Cells transfected with either wild-type NEK10-flag or with T812A-flag mutant were left untreated or stimulated with forskolin (**a**) or isoproterenol (**b**) for 30–60 min. Lysates were immunoblotted for flag and tubulin. **c** Cells were transiently co-transfected with NEK10-flag construct (either wild type or T812A mutant) and HA-ubiquitin. 24 h after transfection, cells were treated with MG132 (20 μM) for 1 h and then stimulated with FSK. Lysates were subjected to immunoprecipitation with anti-flag and immunoblotted with anti-HA and anti-flag. **d** Cells transfected with either wild-type NEK10-flag or with T812A-flag mutant were serum deprived for 36 h, left untreated or stimulated with FSK (3 h) and doubly immunostained for acetylated tubulin and flag. **e** Quantitative analysis of flag-positive ciliated cells expressing either wild type or T812A NEK10 mutant. Cumulative data from three independent experiments are shown. For each group a minimum of 100 cells/experiment was averaged. *$p < 0.05$ versus NEK10-flag and T812A mutant

inhibitor KT5720 followed by FSK or Isoproterenol exposure[31,32]. We confirmed that PKA inhibition prevents the further increase of NEK10 phosphorylation (Fig. 3f). Taken together, these data indicate that PKAc directly phosphorylates NEK10 at T812, both in vitro and in vivo.

We next asked if phosphorylation of T812 by PKA affects NEK10 stability. As suspected, the T812A NEK10 mutant, when expressed in heterologous cells, was not degraded by FSK (Fig. 4a) or isoproterenol (Fig. 4b) treatment, compared to the wild-type protein. These data suggest that modification of NEK10 by PKA is required for proteolysis by the UPS. As shown in Fig. 4c, FSK treatment, indeed, induced poly-ubiquitination of NEK10. Poly-ubiquitination was abrogated by the T812A mutation. Next, we tested whether NEK10 phosphorylation was required for primary cilium disassembly induced by the cAMP cascade. Cells were transiently transfected with NEK10 (either wild type or the NEK10-T812A mutant), serum-deprived for 2 days and then treated with FSK. As shown in Fig. 4d, e, the T812A mutation prevented FSK-induced cilia disassembly, supporting the concept that PKA phosphorylation of T812 primes NEK10 for proteolysis, which results in cilia disassembly.

**NEK10 is required for medaka fish development**. To further prove the role of Nek10 in ciliogenesis, we carried out an in vivo analysis in the medaka fish (*Oryzias latipes*, *ol*) model system by

means of both gene overexpression and knockdown and rescue assays, as already described[33]. *Nek10* was previously reported to be ubiquitously expressed at low levels in most adult mouse tissues[30]. However, its expression during the embryogenesis was not defined. By immunofluorescence analysis, we characterized the expression of NEK10 on medaka embryo at Stage (St.)30 (Supplementary Fig. 10), in which organogenesis is completed. This experiment revealed that *olNek10* is ubiquitously expressed in the whole embryo with high levels in the central nervous system (CNS).

We then analyzed the contribution of NEK10 and cilia disassembly. From St.30–32 onward, overexpression of NEK10-KD, but not wild-type NEK10, led to embryonic morphological abnormalities (Fig. 5a). In particular, embryos overexpressing NEK10-KD showed a dose-dependent phenotype characterized by a significant reduction in trunk size. This defect was associated with microphthalmia, microcephaly, and cardiovascular abnormalities, including pericardial edema. These morphological alterations can be correlated with loss of normal cilia functions, as already reported[34]. In a fraction of the embryos (17%), overexpression of NEK10-KD was embryonic lethal. To better determine the function of NEK10 during embryo development and its role in the onset and progression of the phenotype, we knockdown *olNek10* with a specific morpholino (MO) directed against the second splice donor site (MO-Spl Exo2 NEK10). A significant percentage of these embryos were morphologically

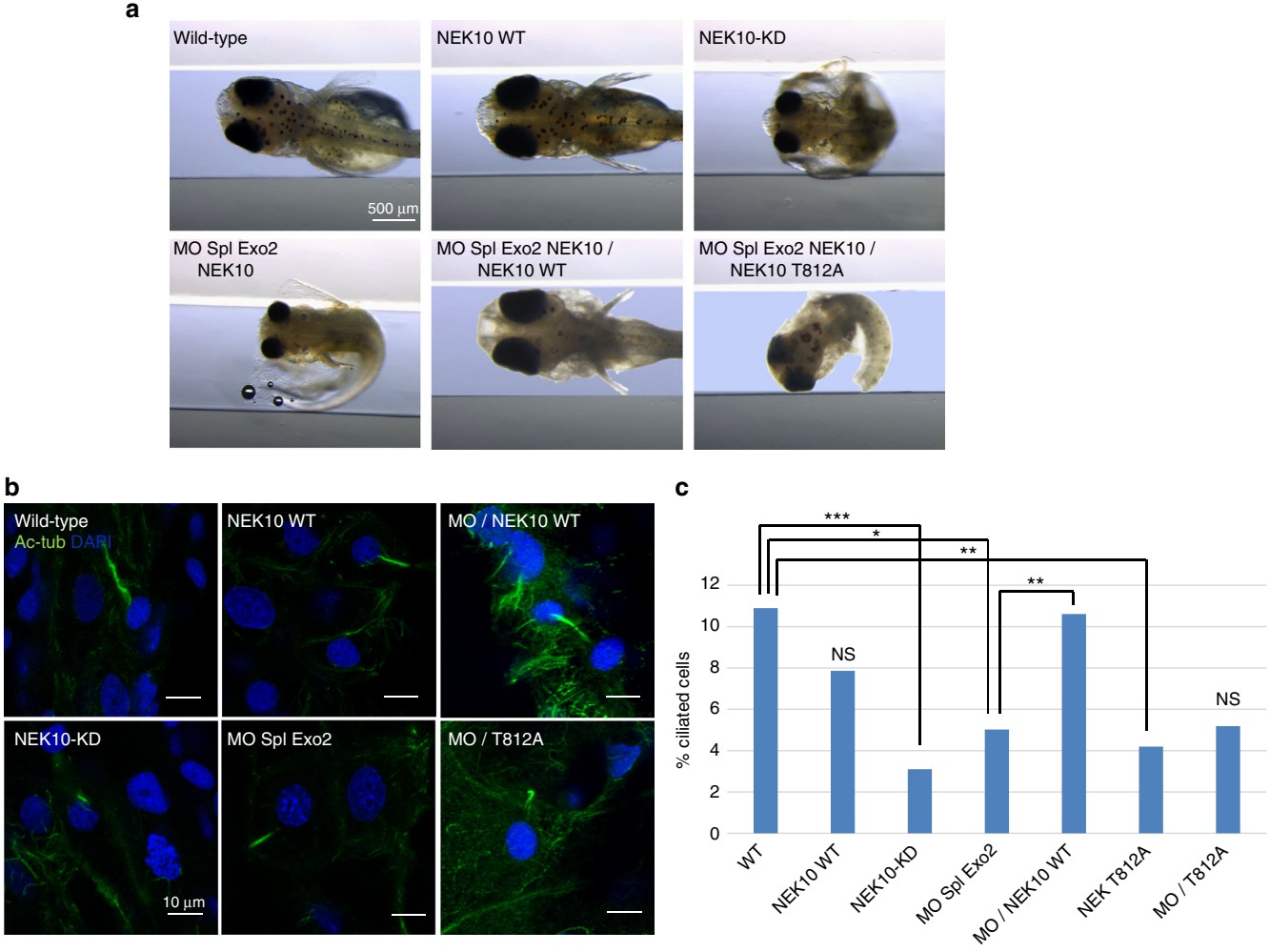

**Fig. 5** NEK10 is required for medaka fish development. **a** Stereomicroscopic images of WT, NEK10 WT, NEK10-KD, MO-Spl Exo2 Nek10, MO-Spl Exo2 Nek10/NEK10 WT, and MO-Spl Exo2 Nek10/NEK10-T812A injected Medaka larvae, at stage 40. NEK10 WT injected larvae appear morphological identical to control WT larvae. NEK10-KD and MO Spl Exo2 show morphological alterations (i.e., microphthalmia, microcephaly, curved body, and others), which were all reminiscent of defective cilia phenotypes in medaka cilia mutants. NEK10-WT co-injections with MO-Spl Exo2 NEK10 rescued the morphological phenotype that was observed in MO-Spl Exo2 NEK10 larvae. NEK10-T812A co-injections with MO-Spl Exo2 NEK10 were not able to rescue the morphological phenotype observed in MO-Spl Exo2 NEK10 larvae. **b** Confocal images of cilia of the neural tube cells in the WT, NEK10 WT, MO-Spl Exo2 NEK10/NEK10 WT, NEK10-KD, MO-Spl Exo2 NEK10, MO-Spl Exo2 NEK10/NEK10-T812A stained with anti-acetylated α-tubulin Ab (green) and DAPI (blue). **c** In the graph is reported cilium length measurements of WT, WT-NEK10, NEK10-KD, MO-Spl Exo2, MO-Spl Exo2/NEK10 WT, NEK10-T812A, and MO-Spl Exo2/NEK10-T812A embryos (n12) (ANOVA test: ***$p < 0.000005$; **$p < 0.00005$; *$p < 0.0005$)

indistinguishable from those overexpressing Nek10-KD and displayed similar defects in the eye, CNS and trunk size, as well as pericardial edema formation. Notably, we did not observe any apparent alteration in left–right asymmetry with respect to controls as assessed by morphological inspection (Fig. 5a). We next asked whether changes in apoptosis and/or cell proliferation were associated with the phenotype observed in morphant embryos. From St.30–32 onward, the terminal deoxynucleotidyl transferase dUTP nick end labeling (TUNEL) assay, a specific method to detect cell death, revealed a significant increase in the number of TUNEL-positive apoptotic cells in the whole embryos in comparison with both control-injected embryos (Supplementary Fig. 10). In particular, we revealed a substantial increase in the number of TUNEL-positive cells in the central nervous system, neuroretina and in the tail of morphant embryos compared to control-injected embryos.

In contrast, we did not observe any alteration in the number of proliferating cells in the whole-morphant embryos in comparison to control-injected embryos, as determined by immunostaining

for phosphorylated histone H3 (PHH3), a specific marker for cells in the M-phase (Supplementary Fig. 10). In agreement to this observation, vibratome sections of PHH3-stained morphant embryos did not reveal any differences in proliferation rate in different regions of morphant embryos, such as eye, brain and tail in comparison to control-injected embryos (Supplementary Fig. 10).

To determine whether both phenotypes (Nek10-KD and MO-Spl Exo2 NEK10) were indeed related to abnormal ciliogenesis, we investigated cilia formation on the apical surface of cells of the neural tube at St.24–26 stage of medaka embryo development using whole-mount immunostaining with anti-acetylated α-tubulin. Notably, a large proportion of embryos injected with NEK10-KD, but not with the of wild-type NEK10, and analysed at St.24–26 (2-days post fertilization, [pf]) showed a significant reduction in cilia length (Fig. 5b, c). Consistent with these observations, MO-Spl Exo2 NEK10 injections induce a statistically significant reduction of cilia length (Fig. 5b, c). Importantly, activation of p53 is an occasional off-target effect of Mo

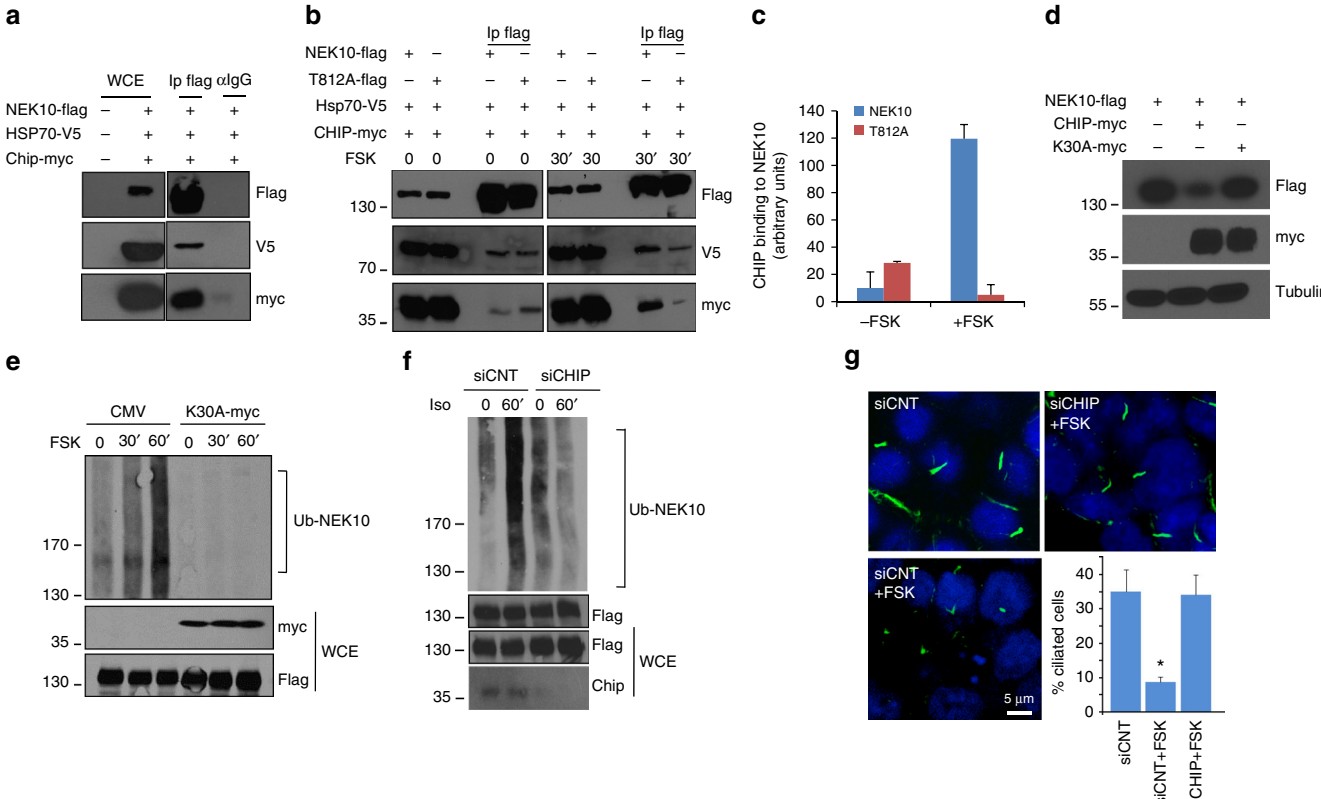

**Fig. 6** CHIP ubiquitylates and degrades NEK10. **a** Cells were co-transfected with NEK10-flag, HSP70-V5 and CHIP-myc. To prevent NEK10 degradation, cells were treated with MG132 (20 µM) for 8 h before collecting. Lysates were immunoprecipitated with anti-flag or with control IgG. The precipitates and lysates were immunoblotted with the indicated antibodies. **b** Cells were co-transfected with NEK10-flag vectors (either wild type or T812A mutant), HSP70-V5 and CHIP-myc, serum deprived for 24 h and left untreated or stimulated with FSK. Lysates were immunoprecipitated with anti-flag. The precipitates and lysates were immunoblotted with the indicated antibodies. **c** Cumulative data of the experiment shown in **b**. The results represent the mean values ± S.E. of three independent experiments. **d** Lysates from cells co-transfected with NEK10-flag and CHIP (either wild type or K30A mutant) were immunoblotted with the indicated antibodies. **e** Cells were co-transfected with HA-ubiquitin, NEK10-flag and CHIP$_{K30A}$-myc or with control plasmid (CMV). 24 h after transfection, to avoid degradation of the ubiquitinated target protein, the cells were treated with MG132 (20 µM) for 8 h and then stimulated with FSK. Lysates were subjected to immunoprecipitation with anti-flag and immunoblotted with anti-HA, anti-flag and anti-myc antibodies. **f** Cells co-transfected with HA-ubiquitin, NEK10-flag and siRNAs (either control siRNA or siCHIP) were serum-deprived overnight and stimulated with isoproterenol. Lysates were subjected to immuno-precipitations with anti-flag antibody. Ubiquitinated NEK10 was revealed by immunoblot with anti-HA antibodies. **g** Cells were transiently transfected with control or with siRNAs targeting CHIP, serum-deprived for 36 h and then left untreated or stimulated with FSK. Primary cilia were visualized by immunostaining for acetylated tubulin. Cumulative data from five independent experiments are shown. For each group a minimum of 100 cells/experiment was averaged. *$p < 0.05$ versus siCNT and siCHIP + FSK

injections, which can be counteracted by injection of a morpholino against p53 (Mo-p53), a key protein involved in the apoptotic pathway[35]. Therefore, to rule out possible non-specific effects of MO-Spl Exo2 NEK10, we coinjected it with the Mo-p53. We did not observe any modifications of the phenotype, which supports the high specificity of MO-Spl Exo2 NEK10 phenotype (Supplementary Fig. 10). Consistent to these observations, co-injection of human wild-type NEK10 that is not recognized by the morpholino MO-Spl Exo2 NEK10, induced a statistically significant rescue of cilia length (Fig. 5b, c), which reflected in a fully rescue of medaka embryo development (Fig. 5a). In contrast, co-injection of human mutated NEK10-T812A did not result in a rescue of morphant phenotype. Altogether, these results strongly supported the absence of off-targeting effects and the specificity of the phenotype induced by Nek10 knockdown. Importantly, overexpression of the NEK10-T812A mutant caused a significant reduction of cilia length compared to both control and wild-type NEK10 injected embryos (Fig. 5b, c). Notably, its co-injection with MO-Spl Exo2 NEK10 exacerbated embryonic morphological abnormalities in comparison to NEK10-KD and MO-Spl Exo2 NEK10 injected

embryos (Fig. 5a). However, no additive reduction of cilia length was observed in the NEK10-T812A/MO-Spl Exo2 NEK10 coinjected embryos (Fig. 5b, c). Taken together, these data suggest that NEK10 affects embryogenesis of medaka fish by modulating the ciliogenesis pathway that is counteracted by PKA phosphorylation at T812.

**CHIP is the NEK10 E3 ubiquitin ligase**. We showed above that cAMP elevation induces NEK10 ubiquitination. We returned to the original proteomics dataset of macromolecular PKA complexes and selected the E3 ubiquitin ligase CHIP (C-terminus of HSP70-interacting protein) for further analyses. CHIP was a hit from the processed and published dataset of direct or indirect PKA interactors[13]. CHIP, encoded by the gene *STUB1*, is an E3 ub-ligase that contains a tetra-tricopeptide (TPR) motif tandem repeats at its N-terminus that mediates interaction with HSP70. The CHIP C-terminus includes a U-box domain separated by a charged coiled-coil region. CHIP is expressed in all mammalian tissues and ubiquitinates most HSP70-bound substrates, acting as a major protein quality system[36–42]. First, we tested whether CHIP interacts with NEK10. As shown in Fig. 6a and

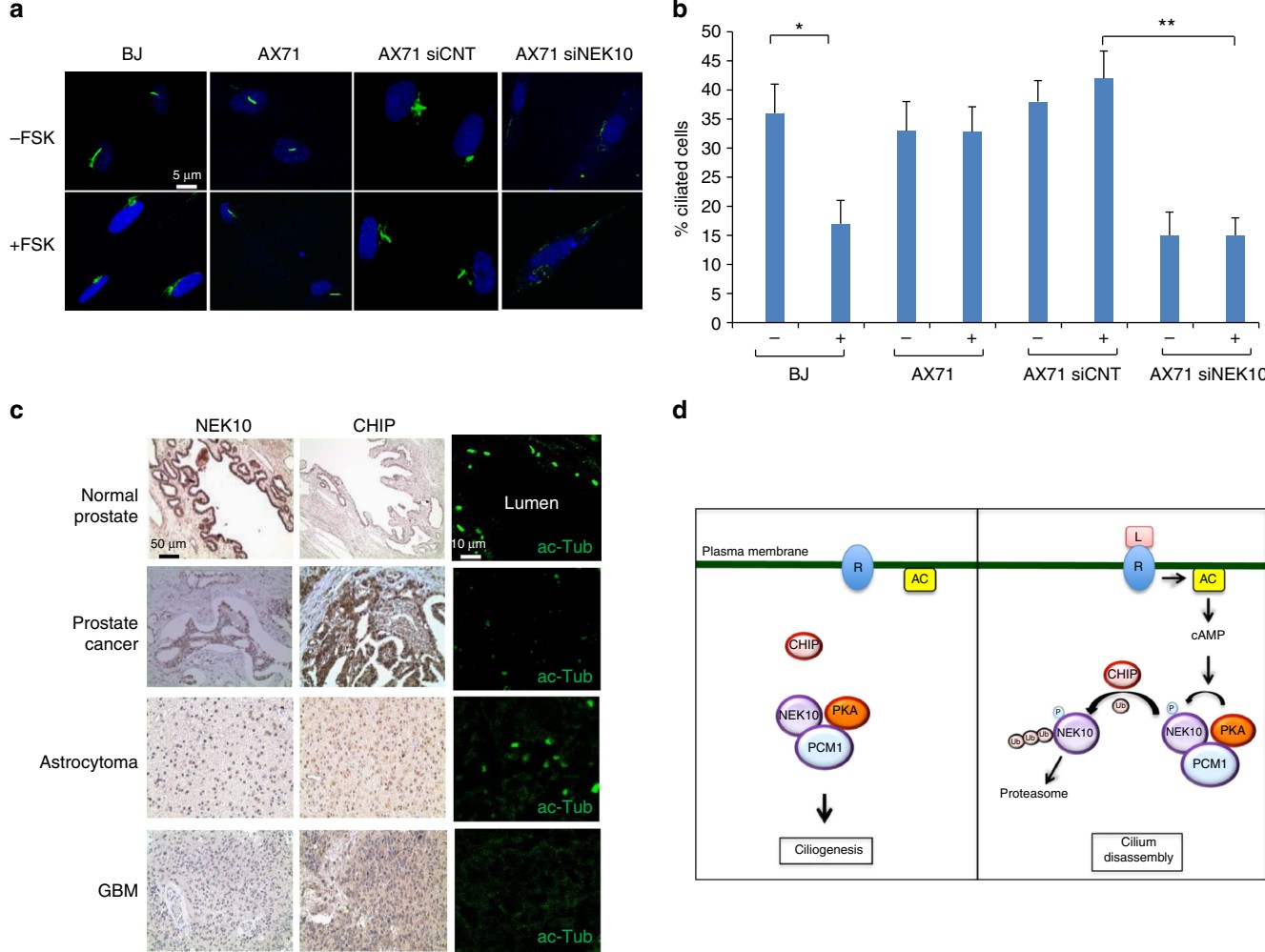

**Fig. 7** CHIP, NEK10 and cilia in SCAR16 fibroblasts and cancer tissues. **a** Skin fibroblasts from healthy volunteers (BJ) and SCAR16 patients (AX71) were serum deprived for 48 h and treated with FSK (80 μM/6 h). Cells were fixed and stained for acetylated tubulin and Draq5. Where indicated, AX71 cells were transiently transfected with siRNA targeting endogenous NEK10, before stimulation. **b** Cumulative data from four independent experiments are shown. For each group a minimum of 25 cells/experiment was averaged. *$p < 0.05$; **$p < 0.01$. **c** Immunohistochemistry for NEK10, CHIP, and acetylated tubulin of the following tissue sections: prostate (normal tissue and high-grade cancer) and glial tumors (astrocytoma and glioblastoma). A total of 8 prostate and 6 glial tumor tissue speciments were analyzed. **d** Schematic depiction of the proposed model: In growth-arrested cells, NEK10 positively contributes to ciliogenesis. Stimulation of adenylate cyclase (AC) by e.g., a defined GPCR ligand increases cAMP levels and activates PKA. PKA phosphorylates NEK10 at T812. Phosphorylated NEK10 undergoes CHIP-mediated ubiquitination and proteolysis, which subsequently leads to cilium disassembly

Supplementary Fig. 11a and Fig. 11b, a stable complex between NEK10, CHIP, and HSP70 could be isolated from cell lysates. The interaction between the three proteins was regulated by cAMP. Thus, treatment with FSK-induced binding between NEK10 and endogenous or exogenous HSP70 and CHIP (Fig. 6b and Supplementary Fig. 11a–c). To note that induction of CHIP binding to NEK10 by FSK was greater compared to that of HSP70, suggesting that Hsp70-independent mechanism(s) could also contribute to CHIP binding to NEK10 (Supplementary Fig. 11c). In contrast, the T812A mutation significantly decreased NEK10 binding to HSP70 and CHIP (Fig. 6b, c). We then asked if CHIP degrades NEK10 in the absence of MG132. As suspected, wild-type CHIP, but not its catalytically inactive mutant (K30A), reduced NEK10 levels (Fig. 6d). Moreover, expression of the CHIP K30A mutant (Fig. 6e) or genetic silencing of endogenous CHIP (Fig. 6f) prevented FSK-induced NEK10 polyubiquitination. These findings supported the idea that cAMP controls NEK10 stability through CHIP. Next, we asked if CHIP mediates the effects of cAMP on cilia stability. Figure 6g,

Supplementary Fig. 11d and Fig. 11e show that downregulation of endogenous CHIP prevented cilia resorption induced by FSK treatment.

**Dysregulation of CHIP affects cilia in human diseases.** Biallelic *STUB1* mutations resulting in aberrant CHIP have been identified in patients with clinical features of autosomal recessive spinocerebellar ataxia-16 (SCAR16). This is a rare genetic syndrome characterized by truncal and limb ataxia resulting in gait instability, mild peripheral sensory neuropathy, and cognitive defects, often as part of a widespread multisystemic neurodegenerative process[43]. Hypogonadism can also be present in these patients (Gordon Holmes syndrome, GHS), consistent with signaling defects and altered responses to hypothalamic hormones[44–47]. Mice lacking *STUB1*/CHIP gene show a phenotype that recapitulates most of the SCAR16 features[48]. Accordingly, we determined whether CHIP mutations affect primary cilia. We analyzed ciliogenesis in primary fibroblasts isolated from cutaneous biopsies of SCAR16 patients or from healthy volunteers.

Figure 7a, b and Supplementary Fig. 12 show that FSK treatment in normal fibroblasts promoted resorption of cilia. In contrast, no major effects of FSK stimulation on cilia were evident in SCAR16 fibroblasts. Interestingly, genetic silencing of NEK10 in SCAR16 fibroblasts markedly reduced the number of ciliated cells, even in the absence of FSK, further supporting a role of the CHIP-NEK10 axis in control of cilium stability.

CHIP is a tumor associated-gene whose levels relate to tumor grade[49,50]. Since CHIP controls NEK10 stability, we explored how CHIP overexpression in cancer tissues is linked to NEK10 levels and ciliogenesis. Strong CHIP staining was evident in high-grade glial tumors (glioblastoma), compared to low-grade glioma (astrocytoma). Similarly, CHIP staining was robust in prostate cancer tissue, compared to a normal counterpart (Fig. 7c). High levels of CHIP staining in malignant tumors were linked to low levels of NEK10, in both prostate and glioma tissues. Moreover, no cilia could be found in glioblastoma samples, compared to astrocytoma (Fig. 7c). Similarly, high-grade prostate cancer tissues show a drastic reduction of ciliated cells, compared to a normal couterpart, most likely reflecting a high rate of cell proliferation, as shown by loss-of-cilia in Ki67-positive cancer cells (Supplementary Fig. 13).

## Discussion

Here we report the identification of a pericentriolar scaffold complex functionally nucleated by PCM1 that controls cAMP signaling events at the primary cilium. In growth-arrested cells, NEK10 is required for cilia assembly. GPCR-mediated activation of PKA induces ubiquitination and proteolysis of NEK10, which culminates in cilia resorption. CHIP was identified as the E3 ligase responsible for NEK10 ubiquitination. Inactivating mutations of CHIP, as seen in SCAR16 disease, severely influenced cAMP-induced cilia resoprtion, whereas increased expression of CHIP was linked to reduced NEK10 levels and reduced ciliogenesis in human cancers.

The primary cilium is a compartmentalized hub for signal integration and propagation relevant for many developmental processes. In dividing cells, the transition between centrosome and primary cilium is functionally linked. In mitotic interphase, centrosomes organize the cytoplasmic microtubule network, whereas in mitosis they regulate mitotic spindle dynamics and cytokinesis. In postmitotic cells, the centrosome migrates to the cell surface, and one of the centrioles differentiates into a basal body from which microtubules nucleate to form a primary cilium. In normal proliferating cells, the cilium can be transiently observed in G1 phase, disappearing when the cell enters the cell cycle[51]. A significant fraction of PKA is localized at the base of cilium through interaction with AKAPs, controlling essential aspects of ciliogenesis and the Hedgehog (Hh) pathway[13–16,18]. However, the impact of PKA activation on the turnover of ciliary proteins and its role in primary cilium stability were largely unknown.

We provide evidence that NEK10 is a novel pericentriolar protein required for primary cilium formation. The localization of NEK10 at this site is mediated by its interaction with PCM1, a pericentriolar scaffold protein involved in different aspects of microtubule dynamics, cell division, and ciliogenesis[7]. We demonstrated that NEK10 is essential for ciliogenesis in mammalian cells and that its function is conserved in lower vertebrates. Thus, interfering with NEK10 expression or activity in Medaka fish negatively impacted on ciliogenesis and embryo development. Defects generated by NEK10 downregulation included reduction of trunk size, microphthalmia, microcephaly, cardiovascular abnormalities, and pericardial edema, supporting a major role of this kinase in ciliary processes underlying vertebrate

development. Following GPCR stimulation, NEK10 becomes a target of the cAMP cascade. Once activated by cAMP, PKA phosphorylates NEK10 at T812. Phosphorylation primes NEK10 for ubiquitination and proteolysis. Decreased NEK10 levels leads to cilia resorption. This regulatory system efficiently couples GPCR signaling to cilia disassembly, accounting for a more general role of cAMP in controlling the sensitivity of cilia to hormones that act at the cell membrane (Fig. 7d). Importantly, the PKA acceptor site T812 is not conserved among the other mammalian members of the NEK family, suggesting a non-redundant mode of regulation of NEK10 by the PKA pathway. By proteomic screening, we identified CHIP as the E3 ligase that ubiquitylates and degrades NEK10, promoting cilia disassembly in response to cAMP stimulation. These findings point to CHIP as a novel regulator of protein turnover at ciliary sites that efficiently couples GPCR signaling to cilia dynamics. This mode of regulation was further supported by evidence that germline inactivating mutations of CHIP that cause SCAR16 disease prevented cAMP-induced disassembly of cilia. Conversely, accumulation of CHIP in malignant tumor lesions was linked to low levels of NEK10 and reduced numbers of ciliated cells.

These findings elucidate the mechanism(s) underlying cilia resorption during GPCR stimulation, both in healthy and disease conditions. They also provide mechanistic insights into how cAMP controls cell growth. It is well established that the cAMP cascade regulates growth and differentiation of a wide variety of cell types. PKA activation can either induce or inhibit cell growth, depending on cell type, or metabolic conditions[52,53]. In growth-arrested endocrine cells, the cAMP-PKA pathway promotes the transition from G0 to G1 phase, allowing the cells to progress through the cell cycle. The transition from quiescent to pro-liferative state requires disassembly of the primary cilium[1]. Previous work revealed that activation of cAMP pathway in growth-arrested, serum supplemented confluent cells may support cilio-genesis[54,55]. This apparent discrepancy could not be ascribed to the different cell models used, since we confirmed that in serum supplemented, confluent cells cAMP stimulation had no major impact on cilium stability (Supplementary Fig. 14). These findings suggest that cAMP pathway may have a dual effect on primary cilia depending on how growth arrest is achieved. In the presence of serum, cAMP contributes to primary ciliogenesis induced by cell confluency, while under serum starvation the same messenger promotes cilium disassembly.

Several targets of PKA have been identified and causally linked to induction of cell growth. However, if and how PKA activation modulates the activity of proteins controlling cilia stability in starved cells was largely unexplored. Our findings help define the relevance of PKA pathway in cilia resorption in the course of hormone stimulation. We show that PKA activation by cAMP agonists targets NEK10 for proteolysis through the UPS. The cAMP cascade induces cilia disassembly and promotes entry into the cell cycle by removing the NEK10 pro-ciliogenic kinase. NEK10 thus represents a nodal point in the ciliary compartment where cAMP signaling and the UPS converge and integrate to control essential aspects of cilia dynamics and, most likely, cell growth. Mutations affecting any component of this proteolytic machinery may alter the sensitivity of the cells to hormones or growth factors, profoundly impacting on cell growth and verte-brate development.

In conclusion, we have identified a PCM1-centered multimeric complex that functionally links second messenger signaling (cAMP), kinase activities (PKA, NEK10), and the UPS (CHIP) to cilia dynamics. This mechanism explains how compartmentalized signaling networks regulate cilia formation in both physiological and pathological conditions.

## Methods

**Cell lines and tissues**. Human embryonic kidney cell line (HEK293) and primary skin fibroblasts from SCAR16 patients were cultured in Dulbecco modified Eagle's medium containing 10% fetal bovine serum in an atmosphere of 5% $CO_2$. Bioptic samples of glial tumors surgically removed from patients were provided by Neuromed Institute of Pozzilli, Italy. All patients gave their informed consents.

**Plasmids and transfection**. Vectors encoding for NEK10-flag (wild type and mutants) and PCM1-HA were provided by Dr Stambolic V and Dr Kamiya A., respectively[30,56]. HA-Ub, CHIP-myc (wild type and K30A mutant), HSP70-V5, were provided by Dr Carlomagno F and RIIβ-myc was provided by Dr Ginsberg MH[57]. PCM1 deletion mutants and NEK10 phosphorylation mutants (T223A and T812A) were generated by PCR using specific oligonucleotides. RSV-PKIα was kindly provided by Dr McKnight GS (University of Washington, US)[58]. siRNAs targeting distinct segments of coding regions of NEK10 and CHIP were purchased from IDT and Life technologies. siRNAs were transiently transfected using Lipofectamine 2000 (Invitrogen) at a final concentration of 100 pmol/ml of culture medium. For siRNA experiments, similar data were obtained using a mixture or four or two independent siRNAs. Transfection efficiency was monitored by including a GFP vector in the transfection mixture. The siRNA sequence (IDT) targeting the 3′-UTR (untranslated region) of human NEK10: sense sequence: 5′-CCACAAGACAUUAGUAAAUUUACTT-3′ antisense sequence: 5′-CGGGU-GUUCUGUAAUCAUUUAAAUGAA-3′ or human CHIP sense sequence: 5′-UUACACCAACCGGGCCUUtt-3′; antisense sequence: 5′-CAAGGCCC GGUUGGUGUAAta-3′.

**Antibodies and chemicals**. The following primary antibodies were used: Goat antibody directed against NEK10 (dilutions: 1/500 for immunoblot, 1/100 for immunofluorescence; Santa Cruz Biotechnology, sc133083), rabbit polyclonal directed against PCM1 (dilutions 1/1000 for immunoblot, 1/100 for immunoprecipitation, 1/100 for immunofluorescence; Abcam), rabbit polyclonal directed against PCM1 (dilutions: 1/1000 for immunoblot, 1/100 for immunoprecipitation; Cell Signaling, #5259), mouse monoclonal directed against CHIP (dilutions: 1/1000 for immunoblot, 1/100 for immunofluorescence; Santa Cruz Biotechnology, sc133083), mouse monoclonal directed against tubulin (dilution 1/8000 for immunoblot; Sigma, T6199), mouse monoclonal directed against acethylated tubulin (dilution 1/100 for immunoblot; Sigma,T7451), rabbit polyclonal directed against acetylated tubulin (dilution 1/100 for immunofluorescence; Abcam, ab125356), mouse monoclonal directed against RIIβ (dilutions: 1/2000 for immunoblot, 1/200 for immunoprecipitation; BD Trasduction, 610625), mouse monoclonal directed against flag (dilutions: 1/4000 for immunoblot, 1/200 for immunoprecipitation; Sigma, F3165), mouse monoclonal directed against myc (dilutions: 1/1000 for immunoblot, 1/100 for immunoprecipitation; Sigma M4439), mouse monoclonal directed against HA (dilution 1/1000 for immunoblot; Covance, 901501), HA (dilution 1/1000 for immunoblot; Roche, 11583816001), rabbit polyclonal directed against actin (dilution 1/3000 for immunoblot, Santa Cruz Biotechnology, sc7210), goat polyclonal directed against NEK10 (1/100 for immunofluorescence Santa Cruz Biotechnology #103067). Phospho-PKA substrate polyclonal rabbit antibody against the (K/R)(K/R)X(S*/T*) motif (dilution 1/1000 for inmunoblot, Cell Signalling, 9621). Antibody protein complexes were detected by HRP-conjugated antibodies and ECL (Amersham Pharmacia).

**Immunoprecipitation and pull down assays**. Cells were homogenized and subjected to immunoprecipitation and immunoblot analyses[59]. GST-fusions were expressed and purified from BL21 (DE3) pLysS cells. GST hybrid proteins immobilized on glutathione beads were incubated for 3 h with cell lysates from HEK293 cells transiently expressing flag-NEK10 constructs in lysis buffer (150 mM NaCl, 50 mM Tris-HCl, pH 7.5, 5 mM $MgCl_2$ 5 mM DDT, 1 mM EDTA, 1% Triton X-100) in rotation at 4 °C for 4 h. Pellets were washed four times in lysis buffer supplemented with NaCl (1 M final concentration) and eluted in Laemmli buffer. Eluted samples were size-fractionated on SDS-PAGE and immunoblotted.

**PKA in vitro phosphorylation assay**. NEK10 WT and T812A coding sequences (786–1115 aa; 519–1115 aa, and 1–519 aa) were PCR-amplified and cloned into the pGEX-5X-1 vector (Sigma) as EcoRI/XhoI fragments (primers GST-NEK10_Fw1: 5′-CGCGGGAATTCATGATGAAATATTTAGACAACTTATC-3′; GST-NEK10_Fw2: 5′-CGCGGGAATTCTATGCAATTTTGGATCATCTTGG-3′; GSTNEK10_Fw3: 5′-CGCGGGAATTCATGCCTGATCAAGATAAAAAGGTG-3′; GST-NEK10_Rv1: 5′-CTCGGGCTCGAGTCATCTTTTGGTTGGGGTG-3′; GST-NEK10_Rv2: 5′-CTGGCCTCGAGTCAGTTGCCTATATATTTCAAAGG-3′). GST-NEK10 recombinant proteins were expressed in the *E. coli* strain BL21-DE3-RIL and expression was induced with 0.8 mM isopropyl-b-D- thiogalactopyranoside (IPTG) at 16 °C for 16 h. Cells were collected by centrifugation, resuspended in in PBS/0.5% Triton and lysed at 1300 psi using a French press device. Clarified lysates were subjected to GST purifications using Glutathione-sepharose beads (GE Healthcare) following the supplier's instructions. His6-PKAc expression was carried out in the *E. coli* strain Rosetta pLysS (Novagen) containing the plasmid pET11d-his6-PKAc. Protein expression was induced with 1 mM IPTG for 3 h at 37 °C. Cells were collected by centrifugation and pellets were resuspended in 50 mM

sodium phosphate pH 8.0, 300 mM NaCl and 10 mM Imidazol. Clarified lysates were subjected to Ni-NTA agarose purification (Invitrogen) following manufacturer's instructions[60]. For the phosphorylation reaction, equal amounts of GST-NEK10 protein beads were incubated with recombinant His6-PKAc in phosphorylation buffer (40 mM Tris at pH 7.5, 0.1 mM EGTA, 10 mM ATP and 10 mM $MgCl_2$) at 30 °C for 20 min at 1000 r.p.m. Beads were washed four times with PBS/0.5% Triton, subjected to SDS-PAGE and immunoblotting using an anti-phospho-(K/R)(K/R)X(S*/T*) specific antibody.

**Medaka stocks**. The Cab-strain of wt medaka fish (*Oryzias latipes*) was maintained following standard conditions (i.e., 12 h/12 h dark/light conditions at 27 °C)[61]. Embryos were staged according to the method proposed by Iwamatsu[62]. All studies on fish were conducted in strict accordance with the institutional guidelines for animal research and approved by the Italian Ministry of Health; Department of Public Health, Animal Health, Nutrition, and Food Safety in accordance to the law on animal experimentation (D.Lgs. 26/2014). Furthermore, all animal treatments were reviewed and approved in advance by the Ethics Committee at the TIGEM Institute, (Pozzuoli (NA), Italy.

**Sequence analysis**. The available medaka *olNEK10* genomic sequences were retrieved from public databases (http://genome.ucsc.edu/) and aligned with human *NEK10* transcript (NM_001031741.3) to identify exons on the basis of sequence conservation[63].

**mRNA and MO injection of medaka embryos**. In vitro synthesis of human NEK10, NEK10-KD (lys548arg), and NEK10-T812A mRNAs were performed following manufacture's instruction[64]. NEK10-KD and NEK10-T812A mRNA was injected at 25–75 ng/μl to observe dose-dependent phenotypes in comparison to NEK10 that did not show any phenotype. Selected working concentrations were 75 ng/μl. A morpholino (Mo; Gene Tools LLC, Oregon, USA) was designed against the splicing donor of exon2 (Mo-NEK10: 5′-TGTGTTGCTCTTCACCTCCCGTCTG-3′) of the medaka orthologous of the NEK10 gene (olNEK10) whereas a control Mo (Mo-p53: 5′-CGGGAATCG-CACCGACAACAATACG-3′) was used as a control. The specificity and inhibitory efficiencies of Mo-NEK10 were determined by using opportune controls[65]. Fifty picoliters of Mo-NEK10 solution (~1/10 of the cell volume) was injected at a 0.09 mM concentration into one blastomere at the one/two-cell stage.[66]

**Whole-mount immunostaining**. Whole-mount immunostaining was performed and photographed, as described[67]. Embryos at Stage 24 and Stage 30 were fixed in 4% paraformaldehyde, 2X phosphate-buffered saline (PBS) and 0.1% Tween-20. The fixed embryos were detached from chorion and washed with PTW 1X. Embryos were digested 20 min with 1 μg/ml proteinase K and washed two fold with 2 mg/ml glycine/PTW 1X. The samples were fixed 20 min in 4% paraformaldehyde, 2X phosphate-buffered saline (PBS) and 0.1% Tween-20, washed with PTW 1X and then incubated for 2 h in block (FBS 1%/PTW 1X), at room temperature. The embryos were incubated with mouse anti-acetylated α-tubulin antibody 1:1000 (6-11B-1; Sigma-Aldrich, St Louis, MO, USA), and goat anti-NEK10 1:100 (Santa Cruz Biotechnology #103067), overnight at 4 °C. The samples were washed with PTW 1X, incubated with the secondary antibody, Alexa-488 goat anti-mouse IgG (ThermoFisher) and Alexa-594 rabbit anti-goat IgG (ThermoFisher), then with DAPI (1:500). Finally, the embryos were placed in glycerol 100%.

**Cell proliferation assay**. The proliferation rate was analysed by immunostaining using an anti-phospho-histone H3 monoclonal antibody (1:400; Cell Signaling Technology). A peroxidase-conjugated anti-rabbit antibody (1:200; Vector Laboratories) was used, followed by diaminobenzidine staining. To prevent pigmentation of the RPE, medaka embryos were incubated with phenylthiourea (PTU). Specifically, from stage 19 (1 day post fertilization) onwards medaka embryos were grown in Yamamoto media supplemented with 0.003% 1-phenyl 2-thiourea (PTU; Sigma-Aldrich) to prevent pigment formation[68].

**Detection of apoptotic cell death**. The extent and distribution of apoptotic cell death was determined by TUNEL, using the In Situ Cell Death Detection Kit, POD (Roche), following the manufacturer's protocol. Alternatively, a peroxidase conjugated anti-rabbit antibody (1:200; Vector Laboratories, Burlingame, CA, USA) was used followed by diaminobenzidine staining[25]. To prevent pigmentation of the RPE, medaka embryos were incubated with phenylthiourea (PTU).

**RNA extraction, cDNA synthesis and RT-PCR**. To confirm abberant splicing in olNEK10 transcript, ctrl and MO injected embryos were subjected to a RT-PCR. Therefore, total RNAs from MO-injected and ctrl Medaka embryos, at stage 24, were extracted by using miRNeasy Mini Kit (Qiagen). The first-strand cDNA synthesis was carried out with the QuantiTect Reverse Transcription Kit (Qiagen), using 1 μg of DNA-free RNA in a final volume of 20 μl, according to the manufacturer's instructions. RT-PCR was employed to evaluate the MO-induced aberrant splicing compared to the control. PCR reaction was carried out with AmpliTaq Gold DNA Polymerase (ThermoFisher) in a total volume of 25 μl

containing 1 μl cDNA and 2 μM primers for MO Spl Exo2 (FW: 5′-AGA-CAGCCAACGTGGCGTCA-3′, REV: 5′-CCATGCTGAACAGAAAGCGA-3′). After initial denaturing step of 10′ at 95 °C, cycling steps were as follows: 1′ at 95 °C, 1′ at 56 °C, and 2′ at 72 °C.

**Statistical analysis**. In all experiments the significance of differences between groups was evaluated by One-way ANOVA with Tukey HSD as post hoc multi-comparison test. Error bars indicate SEM.

**Immunofluorescence and confocal analysis**. For immunofluorescence study, cells were plated on poly-L-lysine coated (10 μg/ml) glass coverslips, fixed, and immunostained with the primary antibodies. Where indicated, cells were serum deprived before fixation. The immunoreactive signals were visualized by fluorescent-labeled secondary antibodies. the fluorescent signals were visualized using a Zeiss LSM 510 Meta argon/krypton laser scanning confocal microscope. Quantification of the immunofluorescent signals and correlation (Pearson's) coefficient were calculated by ImageJ software. Confocal images were acquired using a LSM 700 Zeiss confocal microscope (Carl Zeiss International, Germany). Images were captured using the ZEN software (Carl Zeiss International). Analysis of the length of cilia were performed by ImageJ software. For cilia counting, we considered as cutoff for cilia the length ≥1 μm. High-resolution images were acquired with a Zeiss LSM 880 confocal microscope equipped with Airyscan superresolution imaging module, using a ×63/1.40 NA Plan-Apochromat Oil DIC M27 objective lens (Zeiss MicroImaging, Jena, Germany). Z-stacks covering the whole depth of cells with the interval of 0.018 μm were acquired, followed by Airyscan image processing (set at 7) and analyses using ZEN image acquisition and processing software (Zeiss MicroImaging). Maximum intensity projections shown in the figures were also obtained using ZEN software.

**Immunohistochemical analysis**. Tissue sections were de-paraffinized in xylene and rehydrated with graded ethanol. Antigen retrieval was carried out in citrate buffer (pH = 6.0, 12 min, microwave oven). Endogenous peroxidase activity was quenched with 0.3% hydrogen peroxide for 12 min. Non-specific binding sites were blocked with 5% normal horse serum in TBS-Tween (Wash buffer, Dako, Glostrup, Denmark) for 30 min. Sections were incubated with primary antibodies overnight at 4 °C. All sections were visualized using the Liquid DAB Substrate Chromogen System for peroxidase (DakoCytomation) and were counterstained with hema-toxylin, dehydrated, and mounted.

**cAMP precipitation**. HEK293 cells transiently overexpressing PCM1-flag lysed and subjected to precipitations with Rp-8-AHA-cAMP or 8-AHA-cAMP agarose resins (Biolog) for 2 h to precipitate either activated R or inactive R:C holoenzyme complexes. In this assay, binding of R subunits to the resin-coupled cAMP ana-logue results in re-association of PKAc:R subunits. As a control, we added an excess of cAMP (5 mM) to mask the cAMP-binding sites in the R subunits for pre-cipitation. Resin-associated complexes were washed with the lysis buffer and subjected to immunoblot analysis.

**cAMP quantification**. HEK293 stably expressing the beta-2 adrenergic receptor were serum-starved for 16 h. Cells were exposed to treatments with 10% serum, 20 μM Forskolin or 100 nM Isoproterenol (each for 20 min). After washing steps cell lysis was performed with 0.1 M HCl (for 20 min). The 1:2 diluted soluble cell fractions (in 0.1 M HCl) were subjected to cAMP measurements using the Direct cAMP Enzyme Immunoassay Kit (#CA200, Sigma) following the manufacturer´s instructions. cAMP levels were measured by a plate reader analyses at 415 nm (with 595 nm correction). All the samples were run in duplicates from three independent experiments (three biological replicates each).

**Data availability**. All data are available within the article and Supplementary files or available from the authors upon request.

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

## Acknowledgements

This work was supported by a grant from "Associazione Italiana per la Ricerca sul Cancro" (AIRC: IG15264). Special thanks to Dr Antonella Arcella from Neuromed Institute (Pozzilli, Italy) for providing us the GBM samples and Dr Max Gottesman for critical reading of the manuscript. We thank Francesco Salierno for technical assistance at TIGEM Medaka Fish Facility. L.R. was supported by a FIRC-AIRC Fellowship (#19731). E.S. was supported by grants from the Austrian Science Fund (FWF; P27606, P22608, P30441, and SFB-F44). O.T.-Q. was supported by the Alfonso Martín Escudero Foundation. M.S. was supported by the E-RARE-3 Joint Transnational Call grant "Preparing therapies for autosomal recessive ataxias" (PREPARE) (BMBF, 01GM1607) and by a grant of the Else Kröner-Fresenius Stiftung. I.C. was supported by a grant (TMICCBX16TT) from Telethon Foundation. L.I. was supported by a grant from University of Naples Federico II (CUP E62F17000060001). Special thanks to Drs Stambolic V., Kamiya A., Lim C., Carlomagno F. for kindly sharing the vectors.

## Author contributions

M.P., S.S., M.S., L.R., O.T.-Q., R.D.D., D.I., M.S., L.I., V.A.B., C.G., H.H.L. performed the experiments and analyzed the data. I.C., E.S., and A.F. conceived the experiments, supervised, and analyzed the data. M.S. provided SCAR16 fibroblasts and analyzed the data. A.F. wrote the manuscript with contributions from E.S. and I.C.

## Additional information

**Competing interests:** The authors declare no competing interests.

