## [Peer Review File(PDF 762 kb) · Nature Communications]

Reviewer #1 (Remarks to the Author):

Although several signaling pathways have been reported to govern ciliary disassembly, the mechanisms are not fully understood. In this manuscript, the authors found an interesting phenomenon in which ciliary resorption occurs under conditions in which cAMP levels increase. PKA-dependent phosphorylation of Nek10 induced its degradation by the ubiquitin pathway, and loss of Nek10 led to ciliary shortening. This new cascade may underlie a mechanism to regulate ciliogenesis and ciliary disassembly.

While these findings are potentially quite novel and important, in its current form, the work is rather preliminary. Several important concerns were raised by findings in this manuscript as described below, which significantly diminish enthusiasm for the current study, and as it stands, the current manuscript falls short of being convincing on several counts.

1. The authors confuse pericentriolar satellites and pericentriolar matrix. In particular, see Fig. S3. Therefore, a key point regarding localization of these proteins has not been resolved. Whereas pericentrin localizes to the pericentriolar matrix (not the basal body, as stated), PCM1 resides at satellites, and this image does not show such localization. In general, the localization studies throughout the manuscript are not altogether convincing.

2. The function of cAMP in ciliary length control is still controversial, and significantly more work and discussion would be required to reach the depth expected for an article in Nature Communications. Ideally, the authors should provide insights into how Nek10 regulates ciliation by identifying its substrates. Absent this information, at least some additional mechanistic insights should be provided.

3. Studies showing CHIP ubiquitylation of Nek10 (in particular, figures 3e and 4f) are not compelling. Further, the tumor data in Fig. 5 are preliminary, and no quantitation is shown, rendering conclusions unlikely.

4. Importantly, it is essential that the authors discuss inconsistencies between the results in this manuscript and in previous papers (Besschetnova et al., 2010, doi 10.1016/j.cub.2009.11.072; Abdul-Majeed et al., 2012, doi 10.1007/s00018-011-0744-0). These studies showed that increased cAMP levels and subsequent PKA activation induced ciliary elongation in murine and human cells. Thus, the authors will need substantial experimental additions to the manuscript to resolve these apparent contradictions and prevent further obfuscation in the field.

Other points

1. The paper needs additional proofreading for standard English.
2. The authors have omitted several key citations for centriolar satellites and MIB1 protein, another E3 ligase at satellites.

Reviewer #2 (Remarks to the Author):

Main concerns

The authors propose NEK10 as a required component of ciliogenesis machinery and that PKA controls this process through phosphorylation and subsequent ubiquitination and degradation of NEK10. Although this hypothesis is feasible and very attractive, in my opinion the authors do not provide enough evidences to support it. In addition I believe that the quality of the material presented does not satisfy the standards of a journal as Nat Comm.

Fig1. In fig1b the authors perform and IP with HEK293 cell lysates transfected or not with NEK10-flag. Then they blot with PCM1, Flag and PKAR2b. The authors do not state what percentage of the lysate used in the IP represents the WCE, but in any case it is very uncommon that an associated protein, in this case PCM1, concentrates more efficiently than the immunoprecipitated protein (PKAR2b) itself, this would mean that all PKAR2b is associated with PCM1, even more, each PKAR2b molecule should be associated to several PCM1 molecules to justify this result. Fig 1c, in my opinion the term co-localization should be used only when the proteins show a restricted distribution, PKAR2b and NEK10 both present a broad cytoplasmic distribution that only in some occasions is coincident with PCM1, moreover no special concentration is observed for these proteins at the PCM1 expression area. Fig 1e-f just confirms this point. This broad cytoplasmic distribution normally indicates partial or total nonspecific staining. The authors should provide evidences that the antibodies used are really specific for the indicated proteins. With no cellular references is impossible to tell where the dots of NEK10 are located. In any case, the quality of the pictures should be improved. Then pictures in Supplementary Fig S3a show a very different distribution for PKAR2b as compared with Fig 1c, why this discrepancy? the distribution of PKAR2b in Fig S3 is compatible with the Golgi apparatus, appearing in this case as an overexpressed protein. The way in which the authors quantify the effect of siCNT is mostly irregular, what it means diffuse staining? How has the

threshold of diffuseness been stabilised? Again the authors state that “a significant fraction of NEK10 is associated with the base of cilia” and they mention Fig. 1e and Fig.1f as a proof, in my opinion these pictures do not show that a significant fraction of NEK10 is at the base of the cilia.

Fig 2. In this figure, the authors reduce NEK10 expression by siRNA, then they present a bar graph of number of ciliated cells. The pictures in 2a show some cells with elongated cilia in the control and cells with a short cilia in the NEK10-siRNA treated. The cells are only stained with AcTubulin without any cellular landmark, and no transfection reporter, then they restore the cilium formation by adding recombinant NEK10-flag, what is very strange to me is that the recombinant protein restores the ciliar function without any evident ciliar distribution. It makes me think that NEK10 action on ciliogenesis can be due to an effect everywhere else in the cell. I have no reason to doubt about the bona fide of the cell countings presented, but I would really appreciate that the pictures shown to illustrate it were more convincing. Absolutely in accordance with the authors that reducing the levels or activity of NEK10 induce morphology problems in Medaka embryos however the reported effects are too unspecific to be attributed to cilium malfunction. I would appreciate some information concerning at which organ belong the cells shown in fig 2c.

Fig3. In Fig 3b the authors transfect cells with Flag tagged NEK10 either wt or its mutant forms, then they IP with anti-Flag and blot with an antibody raised against a consensus PKA phosphorylation motive. This antibody has very different affinities for the different PKA substrates, thus although the result is very suggestive the authors should provide evidence that it can recognize the T812 motive. Experiments in Fig3c,d clearly show that NEK-10T812A is resistant to Adenylate Cyclase induced degradation. In my opinion the blot shown in Fig3e can be anything but what the authors pretend it is. First, in my experience poly ubiquitinated proteins tend to separate in discrete bands forming a ladder rather than a smear specially when proteasome activity has been inactivated. Second, while the increase of wt NEK10 is logical, I see no reason why the ubiquitination levels of the mutant should decrease with FSK treatment. In my opinion this experiment does not demonstrate NEK10 ubiquitination and neither an FSK regulation of it. Although the mechanism proposed by the authors is not only feasible but also very attractive in my opinion the results presented are not solid enough to support it. If NEK-10 phosphorylation at T812 is really a key factor for cilium assembly control, one would expect that in the same way that phosphorylation at T812 induces cilium disassembly, the absence of phosphorylation should induce an increase in cilium length, in the same way that it has been reported for Adenylate cyclase inhibitors (Ou et al, Experimental Brain Research 2009). In fig 3f the authors show that cells transfected with NEK-10 T812A do not respond to FSK, but no differences compared with wt NEK-10 are observed in basal conditions, in my opinion cilium length rather than number of ciliated cells would have been a more informative parameter to evaluate cilium assembling. In addition, neither Tuson et al, Development 2014 nor Barci et al, JCS 2012, reported a decrease of ciliated cells upon PKA activity manipulation. All this leads to believe that NEK-10 activity can be related to broader cell functions that may end up affecting cilia formation without being directly implicated in this process.

Fig4. At this point I am not convinced that the authors provided evidence supporting that NEK10 is ubiquitinated in response to cAMP elevation or PKA activity, so whether is CHIP the NEK10 E3 ubiquitin ligase or not, in my opinion adds very little to the relevance of the work.

Fig5. CHIP is a ubiquitin ligase that targets a broad spectrum of proteins for degradation. In my opinion although a correlation between high expression of CHIP and the tumour grade may exist, assuming that this is due to CHIP regulation of NEK10 is just unfounded. Moreover, to make a correlation between a protein level or activity with a tumour development or grade requires a large collection of samples with their pertinent controls and the correct statistical analysis. In no case a chosen collection of pictures can be used to establish such a correlation.

Minor points

Reference n 3 is not well assigned.

References are not well edited

The quality of the fluorescence microscopy pictures is very low

The lettering of the western blots is difficult to follow.

Reviewer #3 (Remarks to the Author):

Comments on Porpora et al.

The manuscript describes the dissection of a pathway that links the production of cAMP to the regulation of the primary cilium. The authors conclude that activated PKA phosphorylates NEK10, which recruits the E3 ubiquitin ligase CHIP, that polyubiquitinates NEK10 leading to its degradation. This is a novel conclusion, of clear interest to the field, and mostly well supported by experimental data. However, there are unsatisfactory aspects of the manuscript and the data relating to cancer and SCAR16 patients is preliminary.

I'm not 100% that NEK10 is phosphorylated by PKA directly, and it would be reassuring to show the effect of a PKA inhibitor on NEK10 phosphorylation. I think this because the site is not unique to PKA but could be the target of another AGC kinase or another basophilic kinase such as Aurora-A, which has been shown to promote ciliary disassembly through HDAC6, and NEK10 could be another substrate.

The evidence for complex formation between Hsp70, CHIP and NEK10 is based on overexpressed, transfected constructs. This is not a convincing approach at the best of times, but Hsp70 is a promiscuous binder, so extra care has to be taken. It would be most convincing if the authors could show specific, phosphorylation-dependent interaction in cells, even more so if the complex localised to the basal body region. This could be done using proximity ligation assay. Alternatively, it would be better to show interactions using endogenous proteins by co-IP.

I don't see that "genetic silencing of endogenous CHIP (Fig. 4f) prevented FSK-induced NEK10 polyubiquitination" (Line 236), poly-ubiquitination appears to be maintained. Please could the authors explain further or show clearer data.

The concluding figure 5 shows a very interesting link between CHIP mutations and deregulation of cilia length from FSK treatment and also between NEK10 expression and loss of cilia in cancer samples. As presented, it seems like more of an afterthought than an integrated part of the rest of the paper and both aspects of the study are preliminary. Could the authors take both of these studies a step further? In the case of SCAR16 fibroblasts, is the effect of FSK treatment through NEK10? In the case of the cancer study, the authors should explain the significance of reduced cilia in the cancer samples – does this simply reflect a higher proportion of proliferating cells and, if so, do the authors believe that, in cancers, CHIP overexpression drives cell-cycle progression through NEK10, at least in part?

Could the authors outline a mechanism that explains why the phosphorylation of NEK10 results in recruitment of CHIP/Hsp70, is there a precedent for CHIP substrates being regulated in this way?

What is downstream of NEK10? Could the authors speculate?

There are some technical issues with the figures.

Figure 1.

The co-localisation of NEK10 and RILB with pericentrin in HEK293 cells is not obvious in panel d, and so the argument that "the three proteins can be present within the same intracellular compartment"

is very weak. Indeed, NEK10 appears to be distributed throughout the cytoplasm in a punctate pattern. Are the authors sure that the NEK10 antibody is specific? To give full confidence in the immunofluorescence localisation data, the authors should show the staining in cells treated with siRNA against NEK10. Furthermore, the authors report what happens to the NEK10 localisation and level when cells are treated with FSK or Iso – you would expect a reduction in NEK10 and perhaps a change in localization? The NEK10 antibody used (goat, santa cruz) is not listed on the website of the company, so this is perhaps an error?

Figure 2.

How many cells/cilia were counted for the graphs in panels a and d?

Figure 3.

The cilia in T812A-flag appear abnormally short, is there an effect of this mutation on cilia morphology or are these not representative images?

How many cells/cilia were counted for the graphs in panel g?

Figure 4.

X-axis labels are missing from panel c

How many cells/cilia were counted for the graphs in panel g?

Figure 5.

The schematic could be enhanced by showing the RIIb subunit of PKA and its interactions.

Rebuttal letter

Reviewer #1 (Remarks to the Author):

Although several signaling pathways have been reported to govern ciliary disassembly, the mechanisms are not fully understood. In this manuscript, the authors found an interesting phenomenon in which ciliary resorption occurs under conditions in which cAMP levels increase. PKA-dependent phosphorylation of Nek10 induced its degradation by the ubiquitin pathway, and loss of Nek10 led to ciliary shortening. This new cascade may underlie a mechanism to regulate ciliogenesis and ciliary disassembly. While these findings are potentially quite novel and important, in its current form, the work is rather preliminary. Several important concerns were raised by findings in this manuscript as described below, which significantly diminish enthusiasm for the current study, and as it stands, the current manuscript falls short of being convincing on several counts.

R. Many thanks to the Reviewer in finding our data 'potentially quite novel and important' and giving us his/her insightful comments/suggestions that helped to improved our manuscript.

1. The authors confuse pericentriolar satellites and pericentriolar matrix. In particular, see Fig. S3. Therefore, a key point regarding localization of these proteins has not been resolved. Whereas pericentrin localizes to the pericentriolar matrix (not the basal body, as stated), PCM1 resides at satellites, and this image does not show such localization. In general, the localization studies throughout the manuscript are not altogether convincing.

*R. We apologize for the confusion about pericentriolar satellites and matrix. We made appropriate changes to the text, as suggested. More importantly, we repeated the localization studies of PCM1, NEK10 and PKAR2b and analyzed the results by confocal microscopy at high resolution. The data presented in the **new Fig. 1d** now better support the notion that NEK10 and PKARIIb signals partly colocalized with PCM1 at centriolar satellites.*

2. The function of cAMP in ciliary length control is still controversial, and significantly more work and discussion would be required to reach the depth expected for an article in Nature Communications. Ideally, the authors should provide insights into how Nek10 regulates ciliation by identifying its substrates. Absent this information, at least some additional mechanistic insights should be provided.

R. We attempted to address this interesting point performing a proteome analysis of NEK10 complexes in cAMP stimulated cells. NEK10 (either wild type or NEK10-S812A mutant) was immunoprecipitated from lysates of transiently transfected cells. Bound

proteins were identified by MS/MS analysis. A list of potential interactors/substrates of NEK10 is now provided in the **new Supplementary Table S1**. We isolated centrosomal proteins (CEP170, CEP68) and GAPs (AGAP1, AGAP3) with a high score probability as potential NEK10 interactors/substrates. These data suggest that NEK10 may regulate cytoskeletal events that take place at the centriolar satellites underlying to ciliogenesis and microtubules dynamics. However, as stated in the manuscript, further work is required to substantiate these preliminary data and characterize the relevant NEK10 centriolar substrates (**please, see p6, line 174**).

3. Studies showing CHIP ubiquitylation of Nek10 (in particular, figures 3e and 4f) are not compelling. Further, the tumor data in Fig. 5 are preliminary, and no quantitation is shown, rendering conclusions unlikely.

*R. We have repeated the ubiquitylation experiments of NEK10 and improved the results, as shown in the **new Fig. 4c**. We have also presented a better quality image for **Fig. 6f**. As for the tumor tissues, we have repeated the experiments using more tissue specimens. The number of the tissue analyzed are now indicated in the **legend to Fig. 7c, line 530**). Moreover, proliferative activity (Ki-67 expression) was examined in the diagnostic tissues by immunohistochemistry using the MIB-1 antibody. Immuno-histochemical analysis of Ki-67 index confirmed that proliferating cells in the cancer tissues do not have primary cilium (**new Supplementary Fig. S10**). Nevertheless, if the Reviewer believes that this aspect at its present stage is too preliminary and should not be included in the manuscript, we can proceed accordingly.*

4. Importantly, it is essential that the authors discuss inconsistencies between the results in this manuscript and in previous papers (Besschetnova et al., 2010, doi 10.1016/j.cub.2009.11.072; Abdul-Majeed et al., 2012, doi 10.1007/s00018-011-0744-0). These studies showed that increased cAMP levels and subsequent PKA activation induced ciliary elongation in murine and human cells. Thus, the authors will need substantial experimental additions to the manuscript to resolve these apparent contradictions and prevent further obfuscation in the field.

R. This is an important point raised by the Reviewer. Previous work delineated the role of cAMP as positive regulator of cilium elongation. The apparent discrepancy with our results is mainly due to different experimental conditions used in both studies. In previous papers, primary cilium was induced by cell confluency and cAMP stimulation was achieved in the presence of serum-supplemented medium. It is well known that serum and cAMP may act in synergy but also in antagonistic way on downstream pathways, even within the same cell type. To avoid interference of serum, we decided to perform cAMP stimulation in absence of serum. However, to further support our point, we repeated the experiments on primary cilium using cells undergoing to growth arrest by cell confluency. In agreement with previous studies,

*we found that cAMP stimulation in the presence of serum did not affect cilium stability, as it did under serum starvation. To make a clear point, we have now included these data in the **new Supplementary Fig. S11** and discussed this aspect further in the Discussion (please, **see p12, line 367**).*

Other points

1. The paper needs additional proofreading for standard English.

R. We have made appropriate proofreading of the text.

2. The authors have omitted several key citations for centriolar satellites and MIB1 protein, another E3 ligase at satellites.

R. We apologize for the missing information. We have now included the appropriate citations on MIB1 (Ref.18)

Reviewer #2 (Remarks to the Author):

Main concerns

The authors propose NEK10 as a required component of ciliogenesis machinery and that PKA controls this process through phosphorylation and subsequent ubiquitination and degradation of NEK10. Although this hypothesis is feasible and very attractive, in my opinion the authors do not provide enough evidences to support it. In addition I believe that the quality of the material presented does not satisfy the standards of a journal as Nat Comm.

Fig1. In fig1b the authors perform and IP with HEK293 cell lysates transfected or not with NEK10-flag. Then they blot with PCM1, Flag and PKAR2b. The authors do not state what percentage of the lysate used in the IP represents the WCE, but in any case it is very uncommon that an associated protein, in this case PCM1, concentrates more efficiently than the immunoprecipitated protein (PKAR2b) itself, this would mean that all PKAR2b is associated with PCM1, even more, each PKAR2b molecule should be associated to several PCM1 molecules to justify this result.

*R. We apologize for the confusion. As suggested, we repeated the coIP experiments in a more quantitative way and included the percentage of the input used for the co-immunoprecipitation assays. The results now appear in the **new Fig 1b**.*

Fig 1c, in my opinion the term co-localization should be used only when the proteins show a restricted distribution, PKAR2b and NEK10 both present a broad cytoplasmic distribution that only in some occasions is coincident with PCM1, moreover no special concentration is observed for these proteins at the PCM1 expression area. Fig 1e-f just confirms this point. This broad cytoplasmic distribution normally indicates partial or total nonspecific staining. The authors should provide evidences that the antibodies used are really specific for the indicated proteins. With no cellular references is impossible to tell where the dots of NEK10 are located. In any case, the quality of the pictures should be improved. Then pictures in Supplementary Fig S3a show a very different distribution for PKAR2b as compared with Fig 1c, why this discrepancy? the distribution of PKAR2b in Fig S3 is compatible with the Golgi apparatus, appearing in this case as an overexpressed protein. The way in which the authors quantify the effect of siCNT is mostly irregular, what it means diffuse staining? How has the threshold of diffuseness been stabilised?

*R. We agree with the reviewer suggestions/comments. To address this point, we repeated the localization studies of PCM1, NEK10 and PKAR2b and analyzed the results by confocal microscopy at high resolution. The data presented in the **new Fig. 1d** now better support the notion that NEK10 and PKAR2b signals partly colocalized with PCM1 at centriolar satellites. Now, the PKAR2b staining from the **new Fig. 1d** is more similar to that shown in the **Supplementary Fig. S3**.*

Again the authors state that “a significant fraction of NEK10 is associated with the base of cilia” and they mention Fig. 1e and Fig.1f as a proof, in my opinion these pictures do not show that a significant fraction of NEK10 is at the base of the cilia.

*R. Thanks to the Reviewer for this comment that helped us to improve the quality of the data presented. Thus, we repeated the immunostaining experiments and analyzed the NEK10 localization at the primary cilium using the super-resolution confocal microscope. As shown in the **new Fig. 2a** and in the **Supplementary movie S1**, in serum deprived cells NEK10 localizes at primary cilium and along the axoneme, indicating that NEK10 resides within the organelle.*

Fig 2. In this figure, the authors reduce NEK10 expression by siRNA, then they present a bar graph of number of ciliated cells. The pictures in 2a show some cells with elongated cilia in the control and cells with a short cilia in the NEK10-siRNA treated. The cells are only stained with AcTubulin without any cellular landmark, and no transfection reporter, then they restore the cilium formation by adding recombinant NEK10-flag, what is very strange to me is that the recombinant protein restores the ciliar function without any evident ciliar distribution. It makes me I think that NEK10 action on ciliogenesis can be due to an effect everywhere else in the cell. I have no reason to doubt about the bona fide of the cell countings presented, but I would really appreciate that the pictures shown to illustrate it where more convincing.

R. As pointed by the Reviewer, we expected that exogenous NEK10 should, at least in part, localize at the cilium, as does the endogenous protein. We attempted to address this point and repeated the experiments several times including also cellular landmarks. The results confirmed the essential role of NEK10 in ciliogenesis. However, the immunostaining conditions using anti-flag antibody in presence of overexpressed flag-tagged protein results in a strong fluorescent signal that appears diffusely distributed throughout the cell. For these reasons, the IF data of the epitope-tagged NEK10 can be useful only to identify transfected cells, rather than to support the localization of the exogenous transgene at the cilium, as does the endogenous protein.

Absolutely in accordance with the authors that reducing the levels or activity of NEK10 induce morphology problems in Medaka embryos however the reported effects are too unspecific to be attributed to cilium malfunction. I would appreciate some information concerning at which organ belong the cells shown in fig 2c.

R. We thank the Reviewer for the recognition of our study in medaka fish, and apologize for not having improved upon the controls and specificity of observed phenotype previously. Therefore, we considerably expanded the revised manuscript with additional results showed in Figure 5. To try to avoid any confusion, when

describing the cilia phenotype in medaka, we have also indicated the tissue of cilia measurement (i.e. neural tube) during embryo development. Notably, using both antisense morpholino oligonucleotide (MO) approach and overexpression NEK10 mutated approaches to inhibit cilia formation during early development, we observed that injected embryos developed through embryonic and early larval stages, but manifested a variety of abnormalities, including microcephaly, curved body, and others, which were all reminiscent of phenotypes in zebrafish and medaka mutants and morphants with defective cilia (Clément et al. 2011 Development. 138,291; Bisgrove et al. 2006 Development 133,4131; Kramer-Zucker et al., 2005 Development 132, 1907; Omran et al. 2008 Nature. 456, 611; Kobayashi et al. 2010 Dev Biol. 347, 62).

Fig3. In Fig 3b the authors transfect cells with Flag tagged NEK10 either wt or its mutant forms, then they IP with anti-Flag and blot with an antibody raised against a consensus PKA phosphorylation motive. This antibody has very different affinities for the different PKA substrates, thus although the result is very suggestive the authors should provide evidence that it can recognize the T812 motive. Experiments in Fig3c,d clearly show that NEK-10T812A is resistant to Adenilate Cyclase induced degradation. In my opinion the blot shown in Fig3e can be anything but what the authors pretend it is.

*R. We have decided to confirm that NEK10 is indeed a new PKA substrate by using two independent approaches (in vivo and in vitro). First, we have expressed a collection of hybrid proteins in bacteria. Then we subjected them to in vitro phosphorylation assays which indeed confirm that T812 is phosphorylated by recombinant PKAc in vitro. Second, we used the specific PKAc inhibitor KT5720. Following PKAc inhibitor treatment we observed that the observed Iso and FSK dependent elevation of NEK10-T812 phosphorylation can be prevented. We present the data in the **new Fig. 3** and in **new Fig. S7**.*

First, in my experience poly ubiquitinated proteins tend to separate in discrete bands forming a ladder rather than an smear specially when proteasome activity has been inactivated. Second, while the increase of wt NEK10 is logical, I see no reason why the ubiquitination levels of the mutant should decrease with FSK treatment. In my opinion this experiment does not demonstrate NEK10 ubiquitination and neither an FSK regulation of it. Although the mechanism proposed by the authors is not only feasible but also very attractive in my opinion the results presented are no solid enough to support it.

R. We understand the point raised by the Reviewer. However, we wish to note that the laddering of polyubiquitylated proteins is more evident when the target protein has a low molecular weight. Nevertheless, to improve the quality of the data presented, we decided to repeat the ubiquitylation assays of NEK10 (either wild type

or mutant). The results shown in the **new Fig. 4c** and in the **Fig. 6f** (now showing the full size-image of NEK10 ubiquitylation experiment in CHIP silenced cells) better support the notion that NEK10 in response to FSK stimulation becomes a target of the UPS.

If NEK-10 phosphorylation at T812 is really a key factor for cilium assembly control, one would expect that in the same way that phosphorylation at T812 induces cilium disassembly, the absence of phosphorylation should induce an increase in cilium length, in the same way that it has been reported for Adenylate cyclase inhibitors (Ou et al, Experimental Brain Research 2009). In fig 3f the authors show that cells transfected with NEK-10 T812A do not respond to FSK, but no differences compared with wt NEK-10 are observed in basal conditions, in my opinion cilium length rather than number of ciliated cells would have been a more informative parameter to evaluate cilium assembling.

R. As suggested by the Reviewer, we have checked the length of cilium in both experimental groups and we experienced some variability in the results. Therefore, we are not confident that such an effect on cilium length of NEK-10 T812A mutant could be documented. However, we wish to point that NEK10 mutation at the PKA site affects cAMP-induced resorption of primary cilium, with marginal effects on other non-PKA-dependent pathways that are involved in ciliogenesis.

In addition, neither Tuson et al, Development 2014 nor Barci et al, JCS 2012, reported a decrease of ciliated cells upon PKA activity manipulation. All this leads to believe that NEK-10 activity can be related to broader cell functions that may end up affecting cilia formation without being directly implicated in this process.

*R. This is an important point raised by the Reviewer. Previous work delineated the role of cAMP as positive regulator of cilium elongation. The apparent discrepancy with our results is mainly due to different experimental conditions used in both studies. In previous papers, primary cilium was induced by cell confluency and cAMP stimulation was achieved in the presence of serum-supplemented medium. It is well known that serum and cAMP may act in synergy but also in antagonistic way on downstream pathways, even within the same cell type. To avoid interference of serum, we decided to perform cAMP stimulation in absence of serum. However, to further support our point, we repeated the experiments on primary cilium using cells undergoing to growth arrest by cell confluency. In agreement with previous studies, we found that cAMP stimulation in the presence of serum did not affect cilium stability, as it did under serum starvation. To make a clear point, we have now included these data in the **new Supplementary Fig. S11** and discussed this aspect further in the ms (**please, see p12, line 367**).*

Fig4. At this point I am not convinced that the authors provided evidence supporting

that NEK10 is ubiquitinated in response to cAMP elevation or PKA activity, so whether is CHIP the NEK10 E3 ubiquitin ligase or not, in my opinion adds very little to the relevance of the work.

*R. To improve the quality of the data presented, we have repeated the ubiquitylation assays of NEK10 (either wild type or mutant). The results shown in the **new Fig. 4c** and in the **Fig. 6f** (that now includes the full size-image of NEK10 ubiquitylation experiment in CHIP silenced cells) better support the notion that NEK10 in response to FSK stimulation becomes a target of the UPS.*

Fig5. CHIP is a ubiquitin ligase that targets a broad spectrum of proteins for degradation. In my opinion although a correlation between high expression of CHIP and the tumour grade may exist, assuming that this is due to CHIP regulation of NEK10 is just unfounded. Moreover, to make a correlation between a protein level or activity with a tumour development or grade requires a large collection of samples with their pertinent controls and the correct statistical analysis. In no case a chosen collection of pictures can be used to establish such a correlation.

*R. We understand the point raised by the Reviewer. However, we wish to point that the expression analysis in cancer tissues was only aimed to show a possible link existing between CHIP, NEK10 and primary cilia in tumor samples. To further support the findings, we have repeated the experiments using more available tissue specimens (as now indicated in the **legend to new Fig. 7c, line 530**). Moreover, proliferative activity (Ki-67 expression) was examined in the diagnostic tissues by immunohistochemistry using the MIB-1 antibody. Immunohistochemical analysis of the Ki-67 index confirmed that proliferating cells in the tissues analyzed do not have primary cilium (**new Supplementary Fig. S10**). Nevertheless, if the Reviewer believes that this aspect at its present stage is too preliminary and should not be included in the manuscript, we can proceed accordingly.*

Minor points

Reference n 3 is not well assigned.

R: Reference corrected.

References are not well edited

R: References edited

The quality of the fluorescence microscopy pictures is very low

*R: Better quality IF images from new experiments are presented in the **new Fig.1d** and **Fig. 2**)*

The lettering of the western blots is difficult to follow.

R: We made the appropriate changes

Reviewer #3 (Remarks to the Author):

Comments on Porpora et al.

The manuscript describes the dissection of a pathway that links the production of cAMP to the regulation of the primary cilium. The authors conclude that activated PKA phosphorylates NEK10, which recruits the E3 ubiquitin ligase CHIP, that polyubiquitinates NEK10 leading to its degradation. This is a novel conclusion, of clear interest to the field, and mostly well supported by experimental data. However, there are unsatisfactory aspects of the manuscript and the data relating to cancer and SCAR16 patients is preliminary.

R: Many thanks to the reviewer in finding our data as 'novel conclusion, of clear interest to the field, and mostly well supported by experimental data'.

I'm not 100% that NEK10 is phosphorylated by PKA directly, and it would be reassuring to show the effect of a PKA inhibitor on NEK10 phosphorylation. I think this because the site is not unique to PKA but could be the target of another AGC kinase or another basophilic kinase such as Aurora-A, which has been shown to promote ciliary disassembly through HDAC6, and NEK10 could be another substrate.

*R. We have decided to confirm that NEK10 is indeed a new PKA substrate by using two independent approaches (in vivo and in vitro). First, we have expressed a collection of hybrid proteins in bacteria. Then we subjected them to in vitro phosphorylation assays which indeed confirm that T812 is phosphorylated by recombinant PKAc in vitro. Second, we used the specific PKAc inhibitor KT5720. Following PKAc inhibitor treatment we observed that the observed Iso and FSK dependent elevation of NEK10-T812 phosphorylation can be prevented. We present the data in the **new Fig. 3** and in the **new Supplementary Fig. S7**.*

The evidence for complex formation between Hsp70, CHIP and NEK10 is based on overexpressed, transfected constructs. This is not a convincing approach at the best of times, but Hsp70 is a promiscuous binder, so extra care has to be taken. It would be most convincing if the authors could show specific, phosphorylation-dependent interaction in cells, even more so if the complex localised to the basal body region. This could be done using proximity ligation assay. Alternatively, it would be better to show interactions using endogenous proteins by co-IP.

*R: We understand the point raised by the reviewer. We have better addressed this point by looking at endogenous HSP70 and CHIP bound to NEK10. As shown in the **new Supplementary Fig. S8a and Fig. S8b**, NEK10 interacts with endogenous HSP70 and CHIP in a cAMP-inducible manner.*

I don't see that "genetic silencing of endogenous CHIP (Fig. 4f) prevented FSK-induced NEK10 polyubiquitination" (Line 236), poly-ubiquitination appears to be

maintained. Please could the authors explain further or show clearer data.

*R: To improve the quality of the data presented, we have repeated the ubiquitylation assays of NEK10 (either wild type or mutant). The results shown in the **new Fig. 4c** and in the **Fig. 6f** (that now includes the full size-image of NEK10 ubiquitylation experiment in CHIP silenced cells) better support the notion that NEK10 in response to FSK stimulation becomes a target of the UPS.*

The concluding figure 5 shows a very interesting link between CHIP mutations and deregulation of cilia length from FSK treatment and also between NEK10 expression and loss of cilia in cancer samples. As presented, it seems like more of an afterthought than an integrated part of the rest of the paper and both aspects of the study are preliminary. Could the authors take both of these studies a step further? In the case of SCAR16 fibroblasts, is the effect of FSK treatment through NEK10?

*R: We thank the Reviewer for this helpful suggestion. We repeated the experiments on fibroblasts carrying inactivating mutations of CHIP and found that FSK effects on cilium are mediated by NEK10. Thus, downregulation of endogenous NEK10 in mutant fibroblasts restored FSK-induced cilium resorption (please, see the **new Fig.7a** and **Fig.7b**).*

In the case of the cancer study, the authors should explain the significance of reduced cilia in the cancer samples – does this simply reflect a higher proportion of proliferating cells and, if so, do the authors believe that, in cancers, CHIP overexpression drives cell-cycle progression through NEK10, at least in part?

*R: As suggested by the Reviewer, we monitored the presence of cilium in rapidly proliferating tumor cells by immunostaining cells with a marker of cell proliferation (Ki-67). We never seen primary cilium in Ki-67 positive, suggesting that proliferation per se is linked to loss of primary cilium. A representative set of the experiments is shown in **Supplementary Fig. S10**. However, we wish to point that the data would only support the existence of a correlation between CHIP, NEK10 and primary cilium in tumor samples, regardless of the causal effect of CHIP overexpression in NEK10 downregulation and loss of cilium in these tissue. Nevertheless, if the Reviewer believes that this aspect at its present stage is too preliminary and should not be included in the manuscript, we can proceed accordingly.*

Could the authors outline a mechanism that explains why the phosphorylation of NEK10 results in recruitment of CHIP/Hsp70, is these a precedent for CHIP substrates being regulated in this way?

R: This is an important point raised by the Reviewer. It is a general idea that HSP70 recruits CHIP to a given substrate/interactor and to our knowledge no signaling

regulation has been reported for this process, so far. Our data demonstrate for the first time that cAMP stimulation induced the recruitment of HSP70/CHIP complex to NEK10. Phosphorylation of NEK10 is a prerequisite for HSP70 binding. However, whether recruitment of CHIP to HSP70 is a consequence or concomitant to NEK10 binding requires further substantial work.

What is downstream of NEK10? Could the authors speculate?

*R: We attempted to address this interesting point performing a proteome analysis of NEK10 complexes in cAMP stimulated cells. NEK10 (either wild type or NEK10-S812A mutant) was immunoprecipitated from lysates of transiently transfected cells. Bound proteins were identified by MS/MS analysis. A list of potential interactors/substrates of NEK10 is now provided in the **Supplementary Table S1**. We isolated centrosomal proteins (CEP170, CEP68) and GAPs (AGAP1, AGAP3) with a high score probability as NEK10 interactors/substrates. These data suggest that NEK10 may regulate cytoskeletal events that take place at the centriolar satellites underlying to ciliogenesis and microtubules dynamics. However, as stated in the manuscript, further work is required to substantiate these preliminary data and characterize the relevant NEK10 centriolar substrates (**please, see p6, line 174**).*

There are some technical issues with the figures.

Figure 1.

The co-localisation of NEK10 and RIIb with pericentrin in HEK293 cells is not obvious in panel d, and so the argument that “the three proteins can be present within the same intracellular compartment” is very weak. Indeed, NEK10 appears to be distributed throughout the cytoplasm in a punctate pattern. Are the authors sure that the NEK10 antibody is specific? To give full confidence in the immunofluorescence localisation data, the authors should show the staining in cells treated with siRNA against NEK10. Furthermore, the authors report what happens to the NEK10 localisation and level when cells are treated with FSK or Iso – you would expect a reduction in NEK10 and perhaps a change in localization? The NEK10 antibody used (goat, santa cruz) is not listed on the website of the company, so this is perhaps an error?

*R: To address the point raised by the Reviewer, we repeated the localization studies of PCM1, NEK10 and PKAR2b and analyzed the results by confocal microscopy at high resolution. The data presented in the **new Fig. 1d** now better support the notion that NEK10 and PKARIIb signals partly colocalized with PCM1 at centriolar satellites. We have added the missing information about the NEK10 antibody in the **Mat&Meth** section (**please, see p19, line 562**). The specificity of the NEK10 antibody was assessed by performing IF analysis of NEK10 silenced cells. If required, we can add this data in the ms.*

Figure 2.

How many cells/cilia were counted for the graphs in panels a and d?

R: we have added the missing information in the figure legend.

Figure 3.

The cilia in T812A-flag appear abnormally short, is there an effect of this mutation on cilia morphology or are these not representative images?

R: We have carefully checked the cilium morphology in T812A-transfected cells and we didn't find reproducible changes of cilium compared to control cells.

How many cells/cilia were counted for the graphs in panel g?

R: We have added the missing information in the figure legend.

Figure 4.

X-axis labels are missing from panel c

R: Correction made.

How many cells/cilia were counted for the graphs in panel g?

R: We have added the missing information in the figure legend.

Figure 5. The schematic could be enhanced by showing the RIIb subunit of PKA and its interactions.

R: We have made the suggested changes to the model.

Reviewer #1 (Remarks to the Author):

This revision has addressed numerous concerns in the prior review. The most interesting aspect of this study continues to be that NEK10 is a novel regulator of ciliogenesis. However, despite the authors' revisions, in its current form, this manuscript may not have a high impact in the absence of a mechanism indicating how NEK10 regulates ciliogenesis. The major points are described below.

1. First, a physiologically relevant mechanism for cAMP-mediated cilia resorption is lacking. From the data, cilia disassembly was only observed in serum-starved cells. For example, are cAMP levels elevated in serum-restimulated cells? If not, it means that cAMP-induced NEK10 degradation only occurs under artificial conditions.

2. Throughout the manuscript, the authors quantified percentages of ciliated cells. The authors argued that cAMP stimulation or siNEK10 reduced the number of ciliated cells. However, those cells retained short cilia. Have the authors counted such cells as non-ciliated? Can such cells re-enter the cell cycle?

3. The authors showed in Fig. 2a that NEK10 is localized on cilia in serum-starved cells. Since PCM1 is not localized on cilia under any conditions, this result means that in ciliated cells, NEK10 was released from the complex and incorporated into cilia. Therefore, NEK10 substrates may be ciliary proteins. Despite this fact, it is not clear why the authors carried out IP and mass-spec with cAMP-stimulated cells. Were those cells serum starved? A description of the experimental details could not be found in the text, legend, or methods. 15 min incubation is enough for NEK10 to be phosphorylated (Fig. 3c), so the substrates may have dissociated from NEK10 under such conditions.

4. From the IHC image in Fig. 7c, the authors cannot conclude that the strong signal for CHIP is relevant to loss of NEK10 or cilia because high-grade cancer tissues always have many mutations. Rigorous statistical analyses are required for such conclusions. Further, is CHIP overexpression sufficient to induce NEK10 degradation? Is cAMP up-regulation not additionally required in this context?

Reviewer #2 (Remarks to the Author):

In my opinion in this reviewed version of the manuscript the authors mostly addressed all the concerns that I had about the paper, so now I believe the manuscript is ready for publication.

Reviewer #3 (Remarks to the Author):

The authors have addressed my point about the direct phosphorylation of NEK10 by PKA – with one exception. The data in Figure 3f are difficult to interpret because KT5720 is not a selective PKA inhibitor at all – in fact it is a broad spectrum kinase inhibitor based on staurosporine - it also potently inhibits PDK1, Aurora kinases, PIM3 among many others (see *Biochem J.* 2007 408: 297–315 and *Sci Signal* 2008 <https://doi.org/10.1126/scisignal.122re4>). KT5720 must not be referred to as a “specific inhibitor” and its activity against other kinases must be referenced.

The evidence for a complex of NEK10 with Hsp70 and CHIP is still very weak. Notably, the new co-IP experiment shown in Figure S8a/b show a clear interaction with CHIP, but the Hsp70 data is unconvincing. Why would there be a ~100-fold enhancement of the CHIP interaction if there is less than 2x enhancement of Hsp70 interaction? Could this be an Hsp70-independent interaction of CHIP? The authors should clarify this point.

The revised figures 4c and 6f are fine. The study using SCAR16 fibroblasts (Fig. 5) now makes much more sense in the context of the paper.

I think that the data on the expression of CHIP in cancerous vs normal tissue as shown in Figure 7 are interesting, so should remain in the manuscript, but the authors should take care not to infer a direct, mechanistic link between CHIP, NEK10 and cilia in cancer unless they are prepared to convincingly demonstrate the connection, because there are many other functions of CHIP. So they should instead be careful with language (e.g. the final sentence of the first paragraph of the Discussion implies a mechanistic link between NEK10-CHIP-cilia in cancer) and clearly state that their evidence for such a link is circumstantial.

It's good to include the list of potential NEK10 partners, this is very interesting additional data.

The revised figure 1 is much improved. The authors should include the data showing specificity of the NEK10 antibody by IF in silenced cells.

The authors have addressed the minor points relating to figures 2, 3, 4 and 5.

Reviewer #4 (Remarks to the Author):

The authors have done in vivo study using misexpression of NEK10-KD and knockdown of NEK10 in medaka to show that NEK10 is required for development. They show control experiments such as misexpressing NEK10 WT and rescuing NEK10 morphant with NEK10 WT co-injection, which are appropriate for this kind of study.

My concerns about the results presented in Fig. 5 are:

- 1) The authors present images of medaka stage 40 larvae to show the phenotype resulting from NEK10 dysfunction. The larvae actually exhibit developmental anomalies, but it is not clear what is the specific defect leading to these anomalies. The authors describe NEK10 dysfunction results in morphological alterations such as microphthalmia, microcephaly and curved body, which are possibly due to growth retardation.
- 2) The authors show cilia of 'the neural tube cells' in Fig. 5b. Are they the ones at the apical surface of cells in the neural tube? Are the images from stage 40 larvae?
- 3) Is NEK10 protein or mRNA expressed ubiquitously in the medaka embryos? Or is it exclusively expressed in some organ? Is any phenotype observed consistently in the organ where NEK10 is expressed?
- 4) Did the NEK10 dysfunction result in laterality defect such as inversed heart looping and/or cystic formation in kidney?

The authors might need to do "statistical analysis for comparison between multigroups' instead of t-test.

Reviewer #1 (Remarks to the Author):

This revision has addressed numerous concerns in the prior review. The most interesting aspect of this study continues to be that NEK10 is a novel regulator of ciliogenesis. However, despite the authors' revisions, in its current form, this manuscript may not have a high impact in the absence of a mechanism indicating how NEK10 regulates ciliogenesis. The major points are described below.

R: We thank the reviewer for positive and constructive comments that helped us to improve the manuscript and significantly strengthen our conclusions. We have addressed the remaining points and revised the manuscript as follows:

1. First, a physiologically relevant mechanism for cAMP-mediated cilia resorption is lacking. From the data, cilia disassembly was only observed in serum-starved cells. For example, are cAMP levels elevated in serum-restimulated cells? If not, it means that cAMP-induced NEK10 degradation only occurs under artificial conditions.

*R: As suggested by the reviewer, we monitored PKA-mediated phosphorylation of cellular substrates with anti-phosphoPKA substrate antibody (as read-out of PKA activity) and cilium stability in serum-restimulated cells. As shown in the new **Supplementary Fig. 1a-c**, readdition of serum to starved cells activates PKA and promotes cilium disassembly. The effects of serum readdition were similar to those observed in cells treated with the GPCR ligand (isoproterenol) or adenylate cyclase activator (forskolin), further supporting the existence of a physiologically relevant mechanism underlying cilium resorption induced by cAMP-PKA pathway.*

2. Throughout the manuscript, the authors quantified percentages of ciliated cells. The authors argued that cAMP stimulation or siNEK10 reduced the number of ciliated cells. However, those cells retained short cilia. Have the authors counted such cells as non-ciliated? Can such cells re-enter the cell cycle?

*R: We apologize for the missing information. Ciliated cells were considered those cells retaining a cilium with a cut-off length of $\geq 1 \mu\text{m}$. We have now included this information in the text (please, see **p.19, line 566**).*

3. The authors showed in Fig. 2a that NEK10 is localized on cilia in serum-starved cells. Since PCM1 is not localized on cilia under any conditions, this result means that in ciliated cells, NEK10 was released from the complex and incorporated into cilia. Therefore, NEK10 substrates may be ciliary proteins. Despite this fact, it is not clear why the authors carried out IP and mass-spec with cAMP-stimulated cells. Were those cells serum starved? A description of the experimental details could not be found in the text, legend, or methods. 15 min incubation is enough for NEK10 to be phosphorylated (Fig. 3c), so the substrates may have dissociated from NEK10 under such conditions.

R: I think the reviewer is referring to the mass spectrometry analyses of immunoprecipitated NEK10 complexes in Table S1. Originally our aim was to identify dynamic interactors (indirect or direct) of NEK10. Therefore we used wt and the A mutant of the functionally relevant T812 site. The mutation abolishes basal phosphorylation and upon cAMP elevation NEK10 phosphorylation is further elevated (in the presence of serum; as shown in Fig.3). In the IPs from the overexpressed HEK293 cell system we could not detect major differences of interactions with wt and the T812A mutant. That is why we included just a selection of functionally interesting hits from both types of IP experiments. We agree that the ideal experimental setup would be to isolate macromolecular PCM1:NEK10 complexes from the appropriate cell compartment. We have modified the main text

accordingly (please, see p.6, line 175 and legend to Table 1).

4. From the IHC image in Fig. 7c, the authors cannot conclude that the strong signal for CHIP is relevant to loss of NEK10 or cilia because high-grade cancer tissues always have many mutations. Rigorous statistical analyses are required for such conclusions. Further, is CHIP overexpression sufficient to induce NEK10 degradation? Is cAMP up-regulation not additionally required in this context?

R: We agree with the Reviewer that a direct, mechanistic link between CHIP, NEK10 and cilia in cancer is not obvious and other mutations occurring in cancer cells may contribute to NEK degradation and cilia disassembly. We have now modified the text pointing to a link, rather to a causal relationship, between CHIP overexpression and NEK10 decrease and loss of cilia in cancer tissues (please see p.11, line 328 and p.12, line 348).

Reviewer #2 (Remarks to the Author):

In my opinion in this reviewed version of the manuscript the authors mostly addressed all the concerns that I had about the paper, so now I believe the manuscript is ready for publication.

R: We are happy to have addressed satisfactorily all concerns raised and we wish to thank the reviewer for the recommendation to accept our revised manuscript.

Reviewer #3 (Remarks to the Author):

The authors have addressed my point about the direct phosphorylation of NEK10 by PKA – with one exception. The data in Figure 3f are difficult to interpret because KT5720 is not a selective PKA inhibitor at all – in fact it is a broad spectrum kinase inhibitor based on staurosporine - it also potently inhibits PDK1, Aurora kinases, PIM3 among many others (see Biochem J. 2007 408: 297–315 and Sci Signal 2008 <https://doi.org/10.1126/scisignal.122re4>). KT5720 must not be referred to as a “specific inhibitor” and its activity against other kinases must be referenced.

R: We agree with the reviewer that KR5720 is not a very specific PKA inhibitor. We include both citation into the manuscript. However, we would like to point out that we did the experiments using the combination of kinase inhibition (KT5720) and indirect & direct cAMP elevation (Forskolin & isoproterenol; = PKA activation). Together with the in vitro data these results point to a PKA-mediated phosphotransferase reaction. We changed the text in the manuscript accordingly (please, see p.7, line 206) and included the suggested references (ref. 50,51).

The evidence for a complex of NEK10 with Hsp70 and CHIP is still very weak. Notably, the new co-IP experiment shown in Figure S8a/b show a clear interaction with CHIP, but the Hsp70 data is unconvincing. Why would there be a ~100-fold enhancement of the CHIP interaction if there is less than 2x enhancement of Hsp70 interaction? Could this be an Hsp70-independent interaction of CHIP? The authors should clarify this point.

*R: We thank the reviewer with this helpful comment. We have now quantified the CHIP binding to NEK10 complex \pm FSK and included the data in the **Supplementary Fig. 10c**. As reported in the figure, the induction of CHIP binding to NEK10 by cAMP was greater (about 8 fold) than that of HSP70, suggesting that other HSP70-independent interaction may also contribute to CHIP recruitment within NEK10 complex. We have now considered this possibility and revised the text accordingly (please, see p.10, line 297).*

The revised figures 4c and 6f are fine. The study using SCAR16 fibroblasts (Fig. 5) now makes

much more sense in the context of the paper.

R: Many thanks.

I think that the data on the expression of CHIP in cancerous vs normal tissue as shown in Figure 7 are interesting, so should remain in the manuscript, but the authors should take care not to infer a direct, mechanistic link between CHIP, NEK10 and cilia in cancer unless they are prepared to convincingly demonstrate the connection, because there are many other functions of CHIP. So they should instead be careful with language (e.g. the final sentence of the first paragraph of the Discussion implies a mechanistic link between NEK10-CHIP-cilia in cancer) and clearly state that their evidence for such a link is circumstantial.

*R: We agree with the reviewer that a direct, mechanistic link between CHIP, NEK10 and cilia in cancer is not obvious and other mutations occurring in cancer cells may contribute to NEK degradation and cilia disassembly. We have now modified the text pointing to **a link**, rather to a causal relationship, between CHIP overexpression and NEK10 decrease and loss of cilia in cancer tissues (please see **p.11, line 328** and **p.12, line 348**).*

It's good to include the list of potential NEK10 partners, this is very interesting additional data.

*R: We have now included the list of NEK10 partners in the main figure (Please, see **Table 1**).*

The revised figure 1 is much improved. The authors should include the data showing specificity of the NEK10 antibody by IF in silenced cells.

*R: Many thanks for the suggestion. We have included the immunofluorescence data showing the specificity of NEK10 antibody in control and NEK10-silenced cells (please, see the **Supplementary Fig. 6a**), as well as in Madaka Embryos (**Supplementary Fig. 9a**)*

The authors have addressed the minor points relating to figures 2, 3, 4 and 5.

R: Thanks again.

Reviewer #4 (Remarks to the Author):

The authors have done in vivo study using misexpression of NEK10-KD and knockdown of NEK10 in medaka to show that NEK10 is required for development. They show control experiments such as misexpressing NEK10 WT and rescuing NEK10 morphant with NEK10 WT co-injection, which are appropriate for this kind of study.

R: We greatly appreciate the efforts the reviewer went through with his/her support and help for the improvement of our manuscript. We also appreciate the overall reviewer comment for the positive evaluation of the study and for the recognition of appropriate controls we used.

My concerns about the results presented in Fig. 5 are:

1) The authors present images of medaka stage 40 larvae to show the phenotype resulting from NEK10 dysfunction. The larvae actually exhibit developmental anomalies, but it is not clear what is the specific defect leading to these anomalies. The authors describe NEK10 dysfunction results in morphological alterations such as microphthalmia, microcephaly and curved body, which are possibly due to growth retardation.

R: Based on the standard staging procedure (Iwamatsu, Mech Dev 121:605–618 (2004), we did not observe any apparent embryonic developmental delay of injected medaka embryos up to stage St30-

32 (*see figure below for the reviewers only*). To exclude a developmental delay, we now provide documented evidence that the proliferation was not affected in Mo-injected embryos (**Supplementary Fig. 9**). In contrast, from st32 onward, the terminal deoxynucleotidyl transferase dUTP nick end labeling (TUNEL) assay, a specific method to detect cell death, revealed a significant increase in the number of TUNEL-positive apoptotic cells. Both results were reported in the revised manuscript and **new Supplementary Fig. 9**.

Iwamatsu T. Stages of normal development in the medaka *Oryzias latipes*. *Mech Dev*. 2004 Jul;121(7-8):605-18.

2) The authors show cilia of 'the neural tube cells' in Fig. 5b. Are they the ones at the apical surface of cells in the neural tube? Are the images from stage 40 larvae?

R: We are grateful to the Reviewer for noting this issue and apologize because we have created some confusion by referring to "we investigated cilia formation neural tube at early stages of medaka embryos...". In the revised manuscript, we have now replaced this sentence accordingly. The text now reads: "we investigated cilia formation on the apical surface of cells of the neural tube at St24 of medaka embryos....." (please, see p.9, line 258)

3) Is NEK10 protein or mRNA expressed ubiquitously in the medaka embryos? Or is it exclusively expressed in some organ? Is any phenotype observed consistently in the organ where NEK10 is expressed?

*R: Following the Reviewer's request, we added, in the new **Supplementary Fig. 9** the Nek10 protein expression. This analysis revealed the olNek10 to be ubiquitously expressed in the whole embryo with high level in the central nervous system (CNS). We modified the revised manuscript, accordingly (please, see p.8, line 228).*

4) Did the NEK10 dysfunction result in laterality defect such as inversed heart looping and/or cystic formation in kidney?

R: We did not observe any apparent left-right asymmetry defects in heart development and/or cystic formation in the kidney as previously reported for the other members of NIMA-related kinases family (i.e. Nek8 and Nek1).

The authors might need to do "statistical analysis for comparison between multigroups' instead of t-test.

*R: As suggested, we have now made the statistical analysis using One-way ANOVA with Tukey HSD as post hoc multicomparison test. The results are now shown in the new Fig. 5c. We have also included this information in the 'Methods' section (please see **p.19, line 553**).*

Reviewer #4 (Remarks to the Author):

- 1) The authors state that they did not observe any alteration in the number of proliferating retinal cells in morphant embryos in comparison (with) control-injected embryos...'. I wonder if there are any signals of brown spots (stained cells). I don't see any difference between WT and the morphant. I don't understand why they specifically refer to the retinal cells, but I guess that the eye pigment (retinal pigment epithelium, RPE) drew their attention. Is this true? Is the eye pigment the signal of pH3? Did they treat the embryos with phenylthiourea to inhibit melanization of the RPE?
- 2) The authors state that apoptotic cell death was increased in the morphant. The data shown in Supplementary figure 9 are not convincing because the background of staining is higher in the morphant and the positions of the sagittal section are different between WT and the morphant.
- 3) What are the stages of embryos shown in Supplementary figure 9?
- 4) The caption (%Ciliated cells) and the error bars in the former Fig. 5C have disappeared in the revised version.
- 5) As for morpholino experiments, it is generally suspected that morpholino injection can cause embryonic lethality or anomaly due to the toxicity. The authors should describe how much morpholino they injected into the cell (they only state the concentration of the injected solution).

Reviewer #5 (Reviewer 1 replacement; Remarks to the Author):

To reply to the comment 1, I think that the authors should show the concentration of cAMP in the serum-starved cells with and without forskolin, comparing with that in serum-restimulated cells. In addition, the authors should perform the knockdown of A kinase using siRNA.

The authors have satisfactorily responded to the comment 2.

It might be required to perform IP without cAMP-stimulation to resolve the concerns in the comment 3.

Reviewer #3 has also raised the concern same as the comment 4 from reviewer #1

Response to Reviewers' comments:

Reviewer #4 (Remarks to the Author):

1) The authors state that they did not observe any alteration in the number of proliferating retinal cells in morphant embryos in comparison (with) control-injected embryos. I wonder if there are any signals of brown spots (stained cells). I don't see any difference between WT and the morphant. I don't understand why they specifically refer to the retinal cells, but I guess that the eye pigment (retinal pigment epithelium, RPE) drew their attention. Is this true? Is the eye pigment the signal of pH3?

OUR REPLY: Thanks to the Reviewer for this comment that helped us to improve the quality of the data presented. We apologize for the confusion when we specifically referred to the retinal cells, and modified both text and **new supplementary Fig. 9** accordingly. Moreover, we now present in the **new Supplementary Fig. 9** (panels B', B'', C', C'') the results of vibratome sections of the morphant embryos at Stage 30. The revised text now reads (page 9 line 257): "In particular, we revealed a substantial increase in the number of TUNEL-positive cells in the central nervous system, retina and in the tail of morphant embryos compared to control-injected embryos. In contrast, we did not observe any alteration in the number of proliferating cells in the whole morphant embryos in comparison to control-injected embryos, as determined by immunostaining for phosphorylated histone H3 (PHH3), a specific marker for cells in the M-phase (**new Supplementary Fig. 9**). In agreement to this observation, vibratome sections of PHH3-stained morphant embryos did not reveal any differences in proliferation rate in different regions of morphant embryos, such as eye, brain and tail in comparison to control-injected embryos (**new Supplementary Fig. 9**)."

Did they treat the embryos with phenylthiourea to inhibit melanization of the RPE?

OUR REPLY: In the revised version of the manuscript we now provide information on the phenylthiourea treatment. The revised text (Materials and Methods) now reads (page 19 line 551): "To prevent pigmentation of the RPE, medaka embryos were incubated with phenylthiourea (PTU) as described previously."

2) The authors state that apoptotic cell death was increased in the morphant. The data shown in Supplementary figure 9 are not convincing because the background of staining is higher in the morphant and the positions of the sagittal section are different between WT and the morphant.

OUR REPLY: Thanks to the Reviewer for this comment that helped us to improve the quality of both proliferation and TUNEL data presented. We now present in the **new supplementary Fig. 9** (panels B', B'', C', C'') the results of vibratome frontal sections of the morphant embryos at Stage 30 for both TUNEL and proliferation assays as previously mentioned (point 1).

3) What are the stages of embryos shown in Supplementary figure 9?

OUR REPLY: In the revised version of the manuscript we now provide information on the stages of embryos.

4) The caption (%Ciliated cells) and the error bars in the former **Fig. 5c** have disappeared in the revised version.

OUR REPLY. We apologize for the inconvenience generated during the conversion to PDF format. We have now included the caption (%ciliated cells) and the error bars in **Fig.5c**.

5) As for morpholino experiments, it is generally suspected that morpholino injection can cause embryonic lethality or anomaly due to the toxicity. The authors should describe how much morpholino they injected into the cell (they only state the concentration of the injected solution).

OUR REPLY: This is an important point raised by the Reviewer. As described in the text we used all necessary controls suggested in Eisen JS, Smith JC. Development 2008. However, to better describe the absence of off-targeting effects we modified the text accordingly. The revised text in now reads (pages 9): "Importantly, activation of p53 is an occasional off-target effect of Mo injections, which can be counteracted by injection of a morpholino against p53 (Mo-p53), a key protein involved in the apoptotic pathway. Therefore, to ruled out a possible nonspecific effects of MO-Spl Exo2 NEK10, we coinjected it with the Mo-p53. We saw no modifications of the phenotype, which supports the high specificity of MO-Spl Exo2 NEK10 phenotype (data not shown). Consistent to these observations, co-injection of human wild-type NEK10 that is not recognised by the morpholino "MO-Spl Exo2 NEK10", induced a statistically significant rescue of cilia length (**Figs. 5b** and **Fig. 5c**), which reflected in a fully rescue of medaka embryo development (**Fig. 5a**). In contrast, co-injection of human mutated NEK10-T812A did not result in a rescue of morphant phenotype. Altogether, these results strongly supported the absence of off-targeting effects and the specificity of the morpholino-induced Nek10 knockdown."

(Eisen JS, Smith JC. Controlling morpholino experiments: don't stop making antisense. Development. 2008 May;135(10):1735-43).

Reviewer #5 (Reviewer 1 replacement; Remarks to the Author):

To reply to the comment 1, I think that the authors should show the concentration of cAMP in the serum-starved cells with and without forskolin, comparing with that in serum-restimulated cells.

OUR REPLY: We have performed cAMP measurements in the presence and absence of serum. Indeed, we confirmed that cAMP levels are elevated in serum-restimulated cells (**new Supplementary Fig.1d**). We have maximally activated cAMP production using the general cAMP elevating agent Forskolin and using agonists for stably overexpressed β_2 adrenergic receptors. These data validated the observation that serum addition contributes to PKA activities as shown in the PKA phosphorylation assay in **new Supplementary Fig.1e**. This information supports the existence of a physiologically relevant and cAMP-PKA involved mechanism relevant for cilium resorption.

In addition, the authors should perform the knockdown of A kinase using siRNA.

REPLY: To address the concern raised by the reviewer, we repeated the cAMP experiments on primary cilium by inhibiting PKA. The knockdown of endogenous PKA for a long time (twenty-four hours transfection, thirty-six hours starvation and then treatment) often resulted in cell death. This can be explained by the essential role played by PKA in basic aspects of cell physiology. Therefore, to address the issue raised, we decided to repeat the experiments by pharmacologically inhibiting PKA for a shorter time (pretreating the cells with a PKA inhibitor for 30 min before forskolin stimulation). The results shown in the **new Fig. 1a** indicate that PKA, indeed, mediates the effects of forskolin on primary cilium disassembly. This finding is supported by the data reported in the manuscript showing that activation of PKA by forskolin directly phosphorylates NEK10 and that NEK10 phosphorylation is required for primary cilium resorption induced by cAMP stimulation.

The authors have satisfactorily responded to the comment 2.

REPLY: Many thanks.

It might be required to perform IP without cAMP-stimulation to resolve the concerns in the comment 3.

OUR REPLY: These supplementary data set provide a few examples of additional proteins which can be found in complex with NEK10. It was not our intention to characterize the macromolecular assembly of NEK10 under different conditions (cAMP) which would rather detract readers from the main message provided in the main figures. We were asked in the first revision to provide this information to get an idea which additional factors might be involved.

Reviewer #3 has also raised the concern same as the comment 4 from reviewer #1

REPLY: Many thanks.

Reviewer #4 (Remarks to the Author):

The results in suppl Fig. 9 are not yet convincing.

First, it looks to me that PTU treatment was not sufficient in these embryos, so the embryos in suppl Figs. 9b and 9c have melanin pigment in RPE. Thus, they cannot state that 'we revealed a substantial increase in the number of TUNEL-positive cells in retina'. In addition, I guess they haven't counted the number of TUNEL- nor PHH3-positive cells.

In the revised manuscript, they cited a paper (Peluso, I. et al. 2013) for the method of PTU treatment. I found that the paper by Peluso does not describe the method, but cites another paper (Conte, I. et al., 2010, PNAS), which does not describe the method either. It seems that they do not treat the embryos with PTU appropriately. They should show the condition of PTU treatment they used for the experiments here.

In the new suppl Figs. 9b and 9c, they present old images of the whole embryos or sagittal sections with new images of the sections. This is inappropriate. The previous comments of mine on 9c images have not been improved. They should show the images of the whole embryos before sectioning. In addition, the whole embryos in 9c do not look like having been stained with DAB (substrate), which is inconsistent to the figure legend.

Reviewer #5 (Remarks to the Author):

The manuscript has been much improved overall. However, there are still concerns about the selectivity of pharmacological PKA inhibition. If the authors are not able to demonstrate the effects of siRNA for PKA in the ciliogenesis, the authors should demonstrate the effects of overexpression of PKA inhibitor peptide.

Response to Reviewers' comments:

Reviewer #4 (Remarks to the Author):

The results in suppl Fig. 9 are not yet convincing.

First, it looks to me that PTU treatment was not sufficient in these embryos, so the embryos in suppl Figs. 9b and 9c have melanin pigment in RPE. Thus, they cannot state that 'we revealed a substantial increase in the number of TUNEL-positive cells in retina'.

OUR REPLY: *We apologize for this inaccuracy. Accordingly, we have replaced the panels illustrating the latter analyses, which are now included in the **revised Supplementary Figure 10 panel c**. However, the residual pigmentation in the RPE was completely irrelevant for our neuroretinal cell counts. To stress our conclusion on these findings and to avoid any confusion, we modified the text that now reads (page 9 lines 276-278): 'In particular, we revealed a substantial increase in the number of TUNEL-positive cells in the central nervous system, neuroretina and in the tail of morphant embryos compared to control-injected embryos.'*

In addition, I guess they haven't counted the number of TUNEL- nor PHH3-positive cells.

OUR REPLY: *We indeed carried out cell counts on both neuroretina and CNS. Accordingly, we now added the graphs showing the cell counts in both neuroretina and CNS (**revised Supplementary Figure 10**).*

In the revised manuscript, they cited a paper (Peluso, I. et al. 2013) for the method of PTU treatment. I found that the paper by Peluso does not describe the method, but cites another paper (Conte, I. et al., 2010, PNAS), which does not describe the method either. It seems that they do not treat the embryos with PTU appropriately. They should show the condition of PTU treatment they used for the experiments here.

OUR REPLY: *We apologize for this omission and we have now added both the method was used and the correct reference in the revised version of the manuscript (Page 19, lines 582-585): "Specifically, from stage 19 (1 day post-fertilization) onwards medaka embryos were grown in Yamamoto media supplemented with 0.003% 1-phenyl 2-thiourea (PTU; Sigma-Aldrich) to prevent pigment formation as described previously⁸⁶.*

New ref #87. *Del Bene F, Wehman AM, Link BA, Baier H. Regulation of neurogenesis by interkinetic nuclear migration through an apical-basal notch gradient. Cell. 2008 Sep 19;134(6):1055-65.*

In the new suppl Figs. 9b and 9c, they present old images of the whole embryos or sagittal sections with new images of the sections. This is inappropriate. The previous comments of mine on 9c images have not been improved. They should show the images of the whole embryos before sectioning. In addition, the whole embryos in 9c do not look like having been stained with DAB (substrate), which is inconsistent to the figure legend.

OUR REPLY: We apologize for this inaccuracy. Accordingly, we have replaced the panels illustrating the latter analyses, which are now included in the **revised Supplementary Figure 10 panel c**.

Reviewer #5 (Remarks to the Author):

The manuscript has been much improved overall. However, there are still concerns about the selectivity of pharmacological PKA inhibition. If the authors are not able to demonstrate the effects of siRNA for PKA in the ciliogenesis, the authors should demonstrate the effects of overexpression of PKA inhibitor peptide.

REPLY. We thank the Reviewer for the positive comment and for the helpful suggestion. We have now addressed the specificity of the PKA inhibition on primary cilium by overexpressing the PKA inhibitor (PKI). To avoid toxic effects derived from long time (three days) inhibition of PKA, the PKI vector was transfected 24 hours from starvation and the transfection long lasted for only 8 hours. After that, cells were stimulated with forskolin and processed. The identification of PKI transfected cells was monitored using a cotransfected GFP vector. The data reported in the **new Supplementary Fig.2** show that expression of PKI completely abolished the effects of forskolin on primary cilium resorption, supporting the model proposed.